# SOHES: Self-supervised Open-world Hierarchical Entity Segmentation

**Shengcao Cao**[1*]  **Jiuxiang Gu**[2]  **Jason Kuen**[2]  **Hao Tan**[2]  **Ruiyi Zhang**[2]
**Handong Zhao**[2]  **Ani Nenkova**[2]  **Liang-Yan Gui**[1]  **Tong Sun**[2]  **Yu-Xiong Wang**[1]
[1]University of Illinois Urbana-Champaign   [2]Adobe Research
{cao44,lgui,yxw}@illinois.edu
{jigu,kuen,hatan,ruizhang,hazhao,nenkova,tsun}@adobe.com

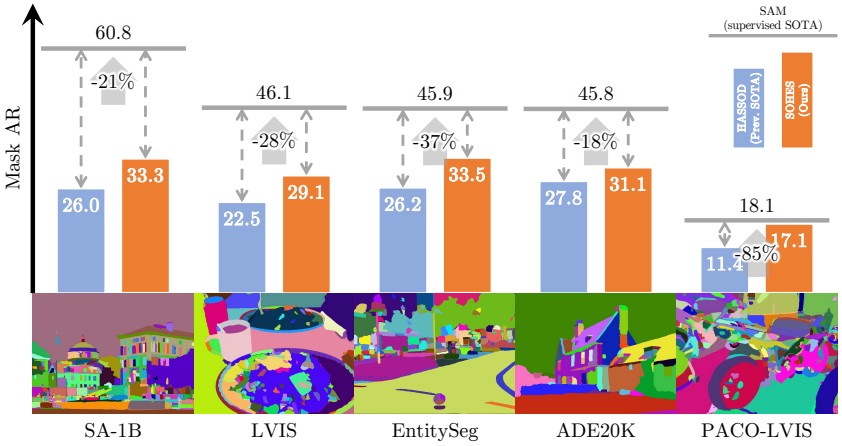

Figure 1: **SOHES boosts open-world entity segmentation with self-supervision on various image datasets.** Compared to prior state of the art, SOHES significantly reduces the gap between self-supervised methods and the supervised Segment Anything Model (SAM) (Kirillov et al., 2023), yet using only 2% unlabeled image data as SAM.

## Abstract

Open-world entity segmentation, as an emerging computer vision task, aims at segmenting entities in images without being restricted by pre-defined classes, offering impressive generalization capabilities on unseen images and concepts. Despite its promise, existing entity segmentation methods like Segment Anything Model (SAM) rely heavily on costly expert annotators. This work presents Self-supervised Open-world Hierarchical Entity Segmentation (SOHES), a novel approach that eliminates the need for human annotations. SOHES operates in three phases: self-exploration, self-instruction, and self-correction. Given a pre-trained self-supervised representation, we produce abundant high-quality pseudo-labels through visual feature clustering. Then, we train a segmentation model on the pseudo-labels, and rectify the noises in pseudo-labels via a teacher-student mutual-learning procedure. Beyond segmenting entities, SOHES also captures their constituent parts, providing a hierarchical understanding of visual entities. Using raw images as the sole training data, our method achieves unprecedented performance in self-supervised open-world segmentation, marking a significant milestone towards high-quality open-world entity segmentation in the absence of human-annotated masks. Project page: https://SOHES.github.io.

## 1 Introduction

Open-world entity segmentation (Qi et al., 2022; 2023) is an emerging vision task for localizing semantically coherent visual entities without the constraints of pre-defined classes. This task, in contrast to traditional segmentation (Long et al., 2015; Chen et al., 2017; He et al., 2017; Kirillov et al., 2019), aims at creating segmentation masks for visual entities inclusive of both "things" (countable

---

*Work done during an internship at Adobe Research.

objects such as persons and cars) and "stuff" (amorphous regions such as sea and sky) (Kirillov et al., 2019; Qi et al., 2022), without regard for class labels. The inherent inclusivity and class-agnostic nature enable open-world entity segmentation to perform strongly on unfamiliar entities from unseen image domains, a frequent real-world challenge in applications such as image editing and robotics. A prominent model for this task is Segment Anything Model (SAM) (Kirillov et al., 2023), which has garnered enthusiastic attention for its impressive performance in open-world segmentation. However, the efficacy of models like SAM depends on the avilability of extensively annotated datasets. To illustrate, SAM is trained on SA-1B (Kirillov et al., 2023), a vast dataset comprising 11 million images and an enormous amount of 1 billion segmentation masks. While automated segmentation plays a central role in building SA-1B, human expertise and manual labor are similarly important, where it takes 14 to 34 seconds to annotate a mask. Meanwhile, it is challenging for human annotators to produce segmentation masks at a consistent granularity, because there is no universally agreed definition of objects and parts. This reliance on intricately annotated datasets and considerable human effort raises a compelling question: *Can we develop a high-quality open-world segmentation model using pure self-supervision?* The prospect of learning from unlabeled raw images without the need for expert annotations is highly appealing.

In fact, self-supervised visual representation learning (Chen et al., 2020; He et al., 2020; Caron et al., 2021; He et al., 2022b) has already shown promise. Such models can effectively exploit useful training signals from purely unlabeled images, resulting in high-quality visual representations that are comparable with those achieved via supervised learning. However, mainstream self-supervised representation learning approaches typically learn holistic representations for whole images, without distinguishing individual entities nor understanding region-level structures. As a result, they cannot be directly used to achieve open-world entity segmentation. Our key insight to bridge this gap is that an intelligent model can *not only learn representations from observations, but can also self-evolve to explore the open world, instruct and generalize itself, continuously refine and correct its predictions in a self-supervised manner*, and ultimately achieve open-world segmentation.

Following this key insight, we propose *Self-supervised Open-world Hierarchical Entity Segmentation (*SOHES*)*, a novel approach consisting of three phases – 1) **Self-exploration:** Starting from a pre-trained self-supervised representation DINO (Caron et al., 2021), we generate initial pseudo-labels to learn from. By clustering visual features based on similarity and locality, we can discern semantically coherent continuous regions that likely represent visually meaningful entities. 2) **Self-instruction:** Our initial pseudo-labels are constrained by the fixed visual representation. To refine the segmentation, we train a Mask2Former (Cheng et al., 2022) segmentation model on the initial pseudo-labels. Even though the initial pseudo-labels are noisy, learning a segmentation model on them can "average out" the noises, thus predicting more accurate masks. 3) **Self-correction:** Building upon these more accurate predictions, we employ a teacher-student mutual-learning framework (Tarvainen & Valpola, 2017; Liu et al., 2020) to further reduce the early-stage noises and adapt the model for open-world segmentation. Throughout the three phases, we rely solely on the raw images, without any human annotations. Equally significantly, due to the compositional nature of things and stuff in natural scenes, our model learns not just to segment entities but also their constituent parts and finer subparts of these parts. During the self-exploration phase, we generate a hierarchical structure of each visual entity from individual parts to the whole. This hierarchical segmentation approach enriches our understanding of visual elements in an open-world context, ensuring a more comprehensive and versatile application.

To summarize, our key contributions include:

- We propose Self-supervised Open-world Hierarchical Entity Segmentation (SOHES) to address the open-world segmentation challenge. We demonstrate the potential of high-quality open-world segmentation by adapting self-supervised representations and learning solely from unlabeled data.
- We develop a method to generate over 100 segmentation masks per image as high-quality pseudo-labels by clustering self-supervised visual features.
- We learn to segment entities and their constituent parts and perform hierarchical association between visual entities. This hierarchical segmentation approach provides a multi-granularity analysis of visual entities in complex scenes.
- We achieve new state-of-the-art performance in self-supervised open-world segmentation, which enhances mask average recall (AR) on various datasets (*e.g.*, improving AR on SA-1B (Kirillov et al., 2023) from 26.0 to 33.3) and closes the performance gap between self-supervised and supervised paradigms, as illustrated in Figure 1.

## 2  RELATED WORK

**Open-world visual recognition.** Open-world recognition (Scheirer et al., 2012; Bendale & Boult, 2015; 2016) aims to recognize and classify visual concepts in an evolving environment where the model encounters unfamiliar objects, which challenges traditional models trained to recognize a fixed set of classes. The task has been extended from classification to detection (Bansal et al., 2018; Dhamija et al., 2020; Joseph et al., 2021; Jaiswal et al., 2021; Kim et al., 2022), segmentation (Hu et al., 2018; Wang et al., 2021; 2022a; Kalluri et al., 2023), and tracking (Liu et al., 2022). In particular, open-world entity segmentation (Qi et al., 2022; 2023) segments entities into semantically meaningful regions without regard for class labels. In this work, we further expand the scope to whole entities and their constituent parts.

**Self-supervised object localization/discovery.** Localizing objects from images in a self-supervised manner requires learning the concept of objects from visual data without any human annotations. Early explorations (Vo et al., 2019; 2020; 2021) formulate an optimization problem on a graph, where the nodes are object proposals (*e.g.*, by selective search (Uijlings et al., 2013)) and the edges are constructed based on visual similarities. Following the observation that the segmentation of the most prominent object can emerge from DINO (Caron et al., 2021), Siméoni et al. (2021; 2023); Wang et al. (2022b) learn object detectors from saliency-based pseudo-labels. Meanwhile, Wang et al. (2022c; 2023) generate pseudo-labels by extending NormCut (Shi & Malik, 2000), and Cao et al. (2023) cluster semantically coherent regions into pseudo-labels. We share a common multi-phase learning paradigm with these prior methods, where pseudo-labels are first discovered from self-supervised representations, and then a detection/segmentation model is learned. However, we contribute *novel and improved designs to each phase*, including 1) a global-to-local clustering algorithm for high-quality pseudo-labeling, 2) a hierarchical relation learning module, and 3) a teacher-student self-correction phase.

## 3  APPROACH

In this section, we first provide an overview of *Self-supervised Open-world Hierarchical Entity Segmentation (*SOHES*)* and then present its three learning phases in the following subsections. Building upon and significantly enhancing the pseudo-label discovery and learning paradigm in prior self-supervised object discovery work (Siméoni et al., 2023; Wang et al., 2023; Cao et al., 2023), SOHES consists of three phases: self-exploration, self-instruction, and self-correction, as shown in Figure 2. 1) In **Phase 1** self-exploration, we start from a pre-trained self-supervised representation DINO (Caron et al., 2021) with a ViT-B/8 (Dosovitskiy et al., 2020) architecture, and initiate our exploration on unlabeled raw images. Our strategy is based on agglomerative clustering (Hastie et al., 2009), and organizes image patches into semantically consistent regions automatically. 2) With these pseudo-labels, we begin **Phase 2** self-instruction. We train a segmentation model composed by a DINO pre-trained ViT backbone (Caron et al., 2021), ViT-Adapter (Chen et al., 2022) (for generating multi-scale features from ViT), and Mask2Former (Cheng et al., 2022) (for the final mask prediction). Through self-instruction, our segmentation model can learn from common visual entities in different images and generalize better than the initial pseudo-labels produced by the frozen ViT backbone. 3) In the final **Phase 3** self-correction, we exploit more self-supervision

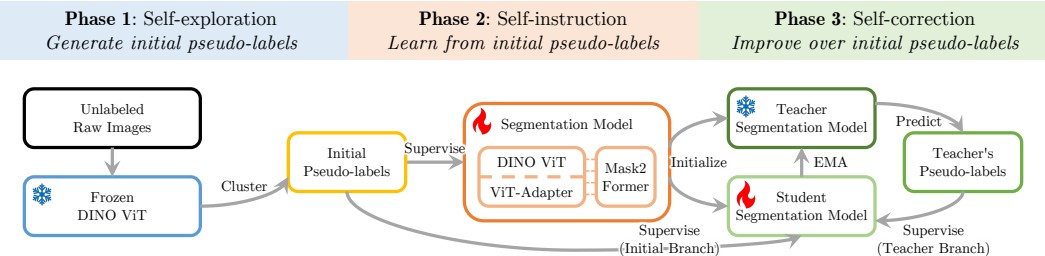

Figure 2: **Three phases of SOHES.** In the first self-exploration phase, we cluster visual features from pre-trained DINO to generate initial pseudo-labels on unlabeled images. Then in the self-instruction phase, a segmentation model learns from the initial pseudo-labels. Finally, in the self-correction phase, we adopt a teacher-student framework to further refine the segmentation model.

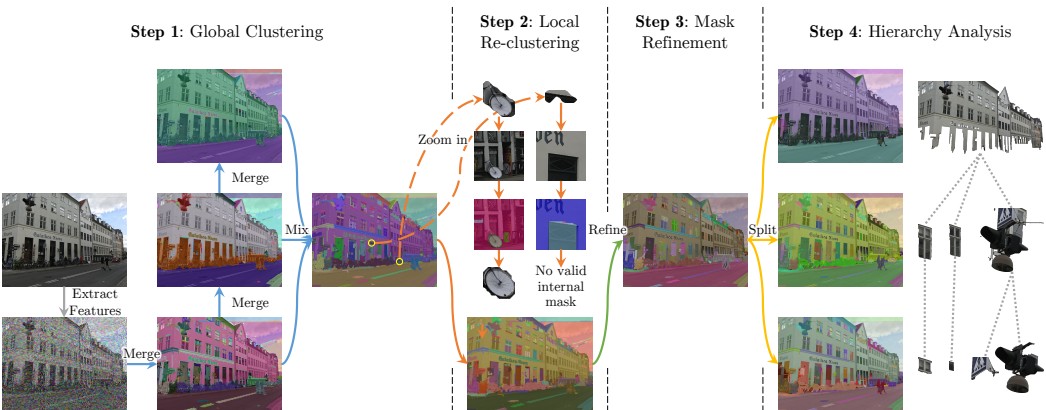

Figure 3: **Self-exploration phase for generating initial pseudo-labels.** This phase consists of four steps. We first merge image patches into regions with high visual feature similarities, then zoom in on the small candidate regions and re-cluster the local images to better discover small entities. After that, we refine the mask details and identify the hierarchical structure among the masks.

signals to lift the limit induced by noises in the initial pseudo-labels. Inspired by semi-supervised learning (Tarvainen & Valpola, 2017; Liu et al., 2020), we employ a teacher-student mutual-learning framework, allowing the student to learn from the improved pseudo-labels generated by the teacher.

## 3.1 SELF-EXPLORATION: GENERATE INITIAL PSEUDO-LABELS

In the self-exploration phase, we generate initial pseudo-labels with several steps delicately designed to include potential entities and their constituent parts of diverse categories. We take a *global-to-local perspective* to first create candidate regions at the global level, and then investigate local images to accurately discover *small* entities. In particular, we begin by clustering patch-level self-supervised features to generate a pool of candidate regions, then filter and refine such candidates into initial pseudo-labeled masks, and finally analyze the hierarchical structure among them. Figure 3 depicts this process with visual examples.

**Step 1** is a global clustering procedure, which merges image patches into semantically meaningful regions. Given an unlabeled image with resolution $S \times S$, we use DINO ViT-B/8 to extract its visual features $\{f_1, \ldots, f_{\frac{S}{8} \times \frac{S}{8}}\}$ corresponding to each $8 \times 8$ patch. Then, we merge these patches in a bottom-up, iterative manner. The initial seed regions are exactly these $8 \times 8$ patches. In each iteration, we find the pair of adjacent regions $(i, j)$ with the highest cosine feature similarity $(f_i \cdot f_j)/(\|f_i\|_2 \cdot \|f_j\|_2)$. These two regions $i$ and $j$ are merged into a new region $k$. The visual feature of the merged region is computed as $f_k = \frac{a_i f_i + a_j f_j}{a_i + a_j}$, where $a_i, a_j$ are the areas of the regions $i, j$. After replacing regions $i$ and $j$ with the new merged region $k$, we continue with the next iteration.

We set a series of merging thresholds $\theta_{\text{merge},1} > \cdots > \theta_{\text{merge},m}$ as criterion for stopping the merging procedure. In general, the highest cosine feature similarity (among all unmerged region pairs) decreases as more regions are merged. When the highest cosine feature similarity goes below one threshold $\theta_{\text{merge},t}$ ($t \in \{1, \ldots, m\}$), we record the merging results that have been obtained so far. Consequently, we can generate $m$ sets of regions, covering various granularity levels. We mix these sets into a pool of regions that may overlap with each other. Non-maximal suppression (NMS) is applied to remove duplicate regions. The thresholds $\{\theta_{\text{merge},t}\}_{t=1}^m$ can be determined based on the desired number of pseudo-labels per image (see Appendix D).

**Step 2** is local re-clustering. In the first step, we have generated a large pool of image regions that may correspond to valid visual entities. However, many small regions tend to be noisy and lack meaningful content. We adopt a global-to-local perspective to re-examine the regions that are smaller than $\theta_{\text{small}}\%$ of the total image area. For each small candidate region, we crop a local image around it, resize it to $S' \times S'$, and re-cluster it with the same procedure as in **Step 1** to obtain subregions of the local crop. Subregions that intersect with the boundaries of the crop are discarded, because they are incomplete within the local crop context. The remaining subregions, along with regions larger than $\theta_{\text{small}}\%$ of the whole image (from **Step 1**), form our initial pseudo-labels. By "zooming in" on the small candidate regions and repeating the clustering procedure at a finer scale, we can better remove noisy pseudo-labels and improve the quality of the remaining ones.

Figure 4: **Ancestor relation prediction in the self-instruction phase.** The prediction target, a binary matrix of ancestor relations, is constructed from the hierarchical structure identified in the self-exploration phase. The ancestor prediction head uses two linear mappings $W_1, W_2$ to transform the query features $Q$ and learns to predict the target ancestors.

In **Step 3**, we leverage the off-the-shelf mask refinement model CascadePSP (Cheng et al., 2020) to further refine the boundaries of the pseduo-label masks. We compute the mask IoUs (intersection-over-union) between the pseudo-labels before and after undergoing the refinement step, and remove the ones that have poor IoUs because they are likely noisy samples.

Finally, **Step 4** focuses on identifying the hierarchical structure embedded within the set of pseudo-labels, which is represented as a forest structure (*i.e.*, set of trees) where the roots are whole entities, and their descendants are parts and subparts, *etc*. We test each pair of pseudo-labels $i$ and $j$ to determine their hierarchical relation: If 1) over $\theta_{\text{cover}}\%$ pixels of pseudo-label $i$ are also in pseudo-label $j$ (meaning that $i$ is covered by $j$), and 2) less than $\theta_{\text{cover}}\%$ pixels of pseudo-label $j$ are in pseudo-label $i$ (meaning that $j$ is larger than $i$), then pseudo-label $j$ is an ancestor of $i$ in the *hierarchy forest*. The smallest ancestor of $i$ is the direct parent of $i$. By testing the pixel coverage between pseudo-labels, we can figure out the hierarchical structure of our pseudo-labels.

## 3.2 Self-instruction: Learn from initial pseudo-labels

In the self-instruction phase, we need to address two problems: 1) The initial pseudo-labels from the previous self-exploration phase contain noises. How to leverage self-supervised learning signals to "average out" the noises? 2) Existing general-purpose segmentation heads cannot predict the hierarchical relations among masks. How to learn the hierarchy forest from the previous phase?

To address the first problem, we train a segmentation model to learn and generalize from the initial pseudo-labels. Through this procedure, the segmentation model can observe valid entities from pseudo-labels which are more frequent than noises, and thus accurately segments unseen images. The model is composed of a ViT-based backbone and a Mask2Former (Cheng et al., 2022) segmentation model. In particular, the backbone is constructed by the same DINO (Caron et al., 2021) pre-trained ViT, and ViT-Adapter (Chen et al., 2022) for producing multi-scale visual feature maps. The ViT backbone is not fixed, and thus we can adapt its features for the segmentation task.

To accomplish the hierarchical segmentation task, we attach a novel *ancestor prediction head* to Mask2Former, which predicts the hierarchical relations among the predicted masks. In parallel to the existing mask and class prediction heads, our ancestor prediction head operates on the query features $Q \in \mathbb{R}^{N \times C}$, where $N$ is the number of queries and $C$ is the query feature dimension. As shown in Figure 4, the learning target of the ancestor prediction is a non-symmetric binary matrix representing the ancestor relations $P \in \{0, 1\}^{N \times N}$, where $P_{i,j} = 1$ represents that mask $i$ is an ancestor of mask $j$, and $P_{i,j} = 0$ otherwise. It is worth noting that a mask $i$ may have no ancestors (as a root in the hierarchy forest), if mask $i$ is a whole entity; mask $i$ may also have more than one ancestor (as a deep node in the forest), if mask $i$ is a part of another part. The ancestor prediction is formulated as:

$$\hat{P} = \text{sigmoid}\left((QW_1)(QW_2)^\top / \sqrt{C}\right) \in \mathbb{R}^{N \times N}, \quad (1)$$

where $W_1, W_2 \in \mathbb{R}^{C \times C}$ are learnable weights for two linear transformations. We use two different linear mappings since the ancestor relations are asymmetric. They are optimized via a binary cross-entropy (BCE) loss $L_{\text{ancestor}} = \text{BCE}(\hat{P}, P)$. At inference time, we can employ topological sorting to reconstruct the forest structure from the binary ancestor relation predictions. Different from prior transformer-based hierarchical segmentation methods like GroupViT (Xu et al., 2022) which are

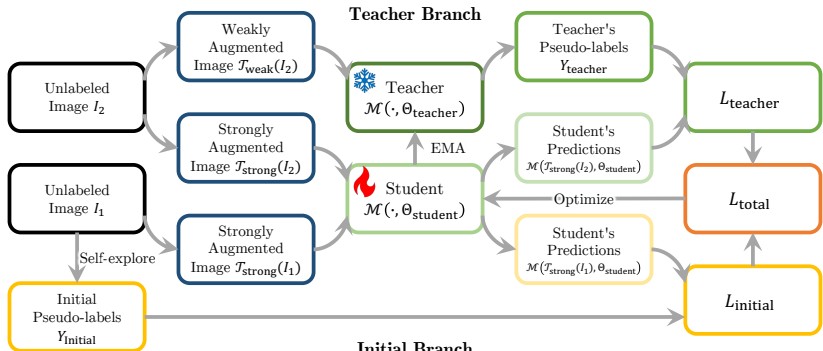

Figure 5: **Teacher-student mutual-learning in the self-correction phase.** We initialize both the teacher and student with the segmentation model learned in the self-instruction phase, which produces better segmentation predictions than the initial pseudo-labels. The student receives supervision from the teacher's pseudo-labels and the initial pseudo-labels. The teacher is updated as the exponential moving average (EMA) of the student.

constrained by the pre-defined number of hierarchical levels, our method is able to predict a variable number of levels and entities.

### 3.3 SELF-CORRECTION: IMPROVE OVER INITIAL PSEUDO-LABELS

Although we have elaborately built a pseudo-labeling process in the self-exploration phase, it is still based on a fixed self-supervised visual representation that is not optimized for image segmentation. Consequently, there may still be initial pseudo-labels that are noisy and can negatively affect our model. Meanwhile, we observe that the segmentation model learned through the self-instruction phase can predict masks that are more reliable and accurate than the clustering results from the self-exploration phase. Motivated by this observation, in the final self-correction phase, we further bootstrap our model by learning from itself and mitigating the impact of noises in the initial pseudo-labels. To achieve self-correction, we adopt a semi-supervised approach that is based on teacher-student mutual-learning (Tarvainen & Valpola, 2017; Liu et al., 2020).

The self-correction phase starts off by initializing two separate segmentation models which are exact clones of the segmentation model produced by the self-instruction phase. We denote one segmentation model as the student model $\mathcal{M}(\cdot, \Theta_{\text{student}})$, which is actively updated through gradient descent; the other segmentation model is the teacher model $\mathcal{M}(\cdot, \Theta_{\text{teacher}})$, which is updated every iteration as an exponential moving average (EMA) of the student: $\Theta_{\text{teacher}} \leftarrow m\Theta_{\text{teacher}} + (1-m)\Theta_{\text{student}}$, where $m \in (0, 1)$ is the momentum. The student receives supervision from both the initial pseudo-labels and the teacher's pseudo-labels. Thus, the total loss is computed as:

$$L_{\text{total}} = L_{\text{seg}}(\mathcal{M}(\mathcal{T}_{\text{strong}}(I_1), \Theta_{\text{student}}), Y_{\text{initial}}) + L_{\text{seg}}(\mathcal{M}(\mathcal{T}_{\text{strong}}(I_2), \Theta_{\text{student}}), Y_{\text{teacher}}), \quad (2)$$

where $I_1$ and $I_2$ are two image batches, $Y_{\text{initial}}$ is the initial pseudo-labels on $I_1$, $Y_{\text{teacher}}$ is the teacher's pseudo-labels by thresholding the predictions $\mathcal{M}(\mathcal{T}_{\text{weak}}(I_2), \Theta_{\text{teacher}})$, $\mathcal{T}_{\text{strong}}$ and $\mathcal{T}_{\text{weak}}$ denote strong and weak data augmentations respectively, and $L_{\text{seg}}$ is the segmentation loss which is composed of a classification loss $L_{\text{cls}}$, a mask prediction loss $L_{\text{mask}}$, and our ancestor prediction loss $L_{\text{ancestor}}$. Figure 5 illustrates the steps in our teacher-student learning approach.

To obtain more reliable supervision from the teacher's predictions $\mathcal{M}(\mathcal{T}_{\text{weak}}(I_2), \Theta_{\text{teacher}})$, we keep only those masks with confidence scores exceeding $\theta_{\text{score}}$ to form the pseudo-labels $Y_{\text{teacher}}$. We observe that the teacher model tends to be less confident when segmenting smaller entities. If the threshold $\theta_{\text{score}}$ is fixed across all masks, it would result in too few pseudo-labels with small areas, and consequently, the student's small entity segmentation performance and overall performance would be impaired. Therefore, we leverage a dynamic threshold:

$$\theta_{\text{score}} = (1 - (1 - a)^\gamma)(\theta_{\text{score, large}} - \theta_{\text{score, small}}) + \theta_{\text{score, small}}, \quad (3)$$

where $a \in (0, 1)$ represents the area ratio of the predicted mask to the whole image, $\gamma > 1$ is a hyper-parameter, and $\theta_{\text{score, small}} < \theta_{\text{score, large}}$ are the pre-defined thresholds for the smallest and largest mask, respectively. This dynamic threshold across different scales allows us to better balance small, medium, and large entities in the teacher's pseudo-labels, and encourages the student model to segment small entities more accurately.

Table 1: **Zero-shot evaluation on various image datasets.** SOHES sets new state-of-the-art self-supervised open-world entity segmentation performance. The collection of the evaluation datasets represents diverse classes in an open world and includes both whole entities and parts. Meanwhile, using just 2% unlabeled images as SAM, SOHES significantly closes the gap between self-supervised methods and the supervised SAM. The evaluation metric is average recall (AR).

| Supervision | Method | Datasets w/ Whole Entities | | | | | Datasets w/ Parts | |
|---|---|---|---|---|---|---|---|---|
| | | COCO | LVIS | ADE | Entity | SA-1B | PtIN | PACO |
| Supervised | SAM (Kirillov et al., 2023) | 49.6 | 46.1 | 45.8 | 45.9 | 60.8 | 28.3 | 18.1 |
| Self-supervised | FreeSOLO (Wang et al., 2022b) | 11.6 | 5.9 | 7.3 | 8.0 | 2.2 | 13.8 | 2.4 |
| | CutLER (Wang et al., 2023) | 28.1 | 20.2 | 26.3 | 23.1 | 17.0 | 28.7 | 8.9 |
| | HASSOD (Cao et al., 2023) | 28.3 | 22.5 | 27.8 | 26.2 | 26.0 | 32.6 | 11.4 |
| | SOHES (Ours) | **30.5** | **29.1** | **31.1** | **33.5** | **33.3** | **36.0** | **17.1** |
| Improvement over HASSOD | | +2.2 | +6.6 | +3.3 | +7.3 | +7.3 | +3.4 | +5.7 |
| Reduced Gap vs. SAM | | -10% | -28% | -18% | -37% | -21% | ∗ | -85% |

∗ SOHES outperforms SAM on PartImagenet.

## 4 EXPERIMENTS

In this section, we thoroughly evaluate SOHES on various datasets and examine the ViT-based backbone improvement for downstream tasks. We perform a series of ablation study experiments to demonstrate the efficacy of modules and steps in SOHES. We also discuss limitations of SOHES in Appendix E. Additional qualitative results are shown in Appendix F.

### 4.1 TRAINING AND EVALUATION DATA

We train our SOHES model on the SA-1B (Kirillov et al., 2023) dataset. In SA-1B, there are 11 million images equally split into 1,000 packs. Unless otherwise specified, we use 20 packs of raw images (2%) for training, and 1 different pack (0.1%) for evaluation.

For evaluation purposes, we test SOHES on various image datasets with segmentation mask annotations in a *zero-shot* manner (*i.e.*, no further training on evaluation datasets). The diversity in the evaluation datasets can simulate the challenge of unseen entity classes and image domains in an open-world setting. Since the annotations in each dataset may only cover entities from a pre-defined list of classes, the commonly used MS-COCO style average precision (AP) metric for closed-world detection/segmentation would incorrectly penalize open-world predictions that cannot be matched with ground truths in known classes. More details of the AP metric are discussed in Appendix B. Following prior work (Kim et al., 2022; Wang et al., 2022a; Liu et al., 2022; Cao et al., 2023), we mainly consider the average recall (AR) metric for up to 1,000 predictions per image when comparing different methods. Other implementation details are in Appendix A.

### 4.2 OPEN-WORLD ENTITY SEGMENTATION

We evaluate SOHES on a variety of datasets, including MS-COCO (Lin et al., 2014), LVIS (Gupta et al., 2019), ADE20K (Zhou et al., 2017), EntitySeg (Qi et al., 2023), and SA-1B (Kirillov et al., 2023). These datasets include natural images of complex scenes, in which multiple visual entities of diverse classes present and are labeled with segmentation masks. Thus, the collection of such evaluation datasets can faithfully reflect the performance of an open-world segmentation model. We compare with recent self-supervised methods FreeSOLO (Wang et al., 2022b), CutLER (Wang et al., 2023), and HASSOD (Cao et al., 2023). We aim to close the gap between self-supervised methods and the supervised state-of-the-art model SAM (Kirillov et al., 2023). *Notably, we use only 2% images as SAM for training* SOHES*, and we do not require any human annotations on these images.*

As summarized in Table 1 and Table 4, SOHES consistently outperforms the prior state-of-the-art HASSOD by large margins (*e.g.*, +7.3 AR on SA-1B and EntitySeg). Meanwhile, SOHES significantly closes the gap between self-supervised methods and supervised methods. For instance, using only 2% unlabeled data in SA-1B, SOHES already achieves over half AR of SAM. SOHES also reduces the gap between self-supervised methods and SAM on SA-1B relatively by 37%.

### 4.3 PART SEGMENTATION

In additional to whole entities, SOHES also learns to segment their constituent parts and subparts. To evaluate our hierarchical segmentation results, we compare them with the ground-truth mask

Figure 6: **Downstream performance of ViT-based backbones.** We freeze the ViT and fine-tune a ViT-Adapter and a lightweight segmentation/detection head on ADE20K/MS-COCO. The backbone further fine-tuned in SOHES is more adapted to dense-prediction tasks.

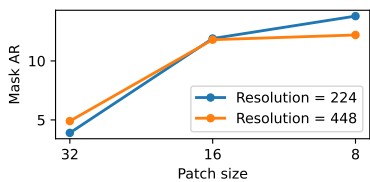

Figure 7: **Mask quality of the initial pseudo-labels produced by DINO backbones with different pre-training configurations.** A small pre-training patch size leads to better fine-grained features and high-quality pseudo-labels.

| Backbone | ADE20K mIoU | MS-COCO AP |
|---|---|---|
| Pre-trained by DINO | 35.2 | 22.7 |
| Fine-tuned by SOHES | **39.6** | **24.4** |

annotations of object parts (*e.g.*, heads and tails of animals) in two datasets, PartImageNet (He et al., 2022a) and PACO-LVIS (Ramanathan et al., 2023), and summarize the results in Table 1 and Table 5. Compared with prior self-supervised baselines, SOHES more accurately localizes meaningful parts of entities, and almost doubles CutLER's performance on PACO (8.9 AR → 17.1 AR). Impressively, SOHES *outperforms SAM* on PartImageNet and is on par with SAM on PACO. The reason is that SOHES can predict more parts and subparts than SAM (see Figure 14), which are the focus of the two datasets' annotations.

## 4.4 Improved backbone features

Through our self-instruction and correction phases, we adapt self-supervised representation DINO to an open-world segmentation model. Consequently, our fine-tuned visual backbone can better fit into other dense prediction tasks. To test such abilities, we compare a) the ViT-B/8 backbone pre-trained by DINO and b) the backbone further tuned in SOHES, in downstream tasks of semantic segmentation on ADE20K and object detection on MS-COCO. The downstream fine-tuning is performed in a minimalistic style, mimicking the linear probing (Chen et al., 2020; He et al., 2020) in self-supervised representation learning. For semantic segmentation, we directly attach a linear classifier on the feature maps from ViT-Adapter; for object detection, we attach the simplest RetinaNet (Lin et al., 2017) detection head on the ViT-Adapter. We keep the ViT parameters frozen during the supervised fine-tuning. Table 6 summarizes the results, demonstrating that SOHES can adapt the ViT-based backbone to generate better features for dense prediction downstream tasks.

## 4.5 Ablation study

In this subsection, we ablate the design choices in SOHES on SA-1B, and provide our insights for future research in self-supervised open-world segmentation. Further details about the choices of hyper-parameters and model architectures are discussed in Appendix D.

**DINO backbone.** In recent self-supervised object localization/discovery work (Siméoni et al., 2023; Wang et al., 2023), researchers prefer the ViT backbone pre-trained by DINO, in particular ViT-B with patch size 8. We have also observed that a ViT backbone with patch size 8 leads to better mask quality in SOHES. To investigate this, we use DINO to pre-train ViT-B backbones with varying patch size and input resolution configurations, with a shorter 100-epoch training schedule (DINO originally pre-trains on ImageNet (Deng et al., 2009) for 300 epochs). Then, we repeat our self-exploration phase with these backbones, and the resulting mask quality comparison is summarized in Figure 7 and Table 7. From this comparison, we can observe that the small patch size is positively correlated with the mask quality. When the patch size decreases from 32 to 8, the AR significantly improves. Meanwhile, the input resolution does not influence the mask quality as much. The small patch size may better support the ViT to capture pixel-aligned details for localizing entites, and thus is more suited in our self-supervised segmentation task. It is worth noting that we cannot further reduce the patch size due to computational constraints. Whenever the patch size is halved, ViT needs to process $4\times$ patches, and perform $16\times$ computation in self-attention. Therefore, the off-the-shelf DINO ViT-B/8 is the best choice in our task.

**Steps in self-exploration.** In our self-exploration phase, we have delicately designed a series of steps to generate, select, and refine the pseudo-labels. We summarize the impact of the design choices in Table 2. In the first global clustering step, if we use one fixed merging threshold $\theta_{\text{merge}}$, a larger $\theta_{\text{merge}}$ leads to more masks per image and better coverage of entities (increasing AR), and also introduces noises (oscillating AP). We choose to mix the results with different thresholds together

Table 2: **Impact of each step in self-exploration.** We mix the global clustering results from multiple merging thresholds, adopt the local re-clustering, and use the off-the-shelf CascadePSP mask refinement module to obtain the best initial pseudo-labels.

| Step | Choice | Masks/Img | Time/Img (sec) | AP | $AR_S$ | $AR_M$ | $AR_L$ | AR |
|---|---|---|---|---|---|---|---|---|
| 1 | $\theta_{\text{merge}} = 0.1$ | 1 | 4.9 | 0.2 | 0.0 | 0.0 | 0.3 | 0.1 |
| | $\theta_{\text{merge}} = 0.2$ | 3 | 4.9 | 0.8 | 0.1 | 0.1 | 0.6 | 0.3 |
| | $\theta_{\text{merge}} = 0.3$ | 9 | 4.5 | 0.9 | 0.4 | 0.6 | 2.0 | 1.0 |
| | $\theta_{\text{merge}} = 0.4$ | 23 | 3.8 | 0.6 | 0.7 | 1.5 | 5.4 | 2.6 |
| | $\theta_{\text{merge}} = 0.5$ | 58 | 2.7 | 1.4 | 1.2 | 3.4 | 11.1 | 5.4 |
| | $\theta_{\text{merge}} = 0.6$ | 131 | 2.2 | 0.6 | 1.5 | 6.0 | 15.3 | 8.1 |
| | Mix w/ NMS | 148 | 5.3 | 1.1 | 1.9 | 6.5 | 17.2 | **9.1** |
| 2 | w/ local re-clustering | 115 | 8.4 | 2.0 | 5.1 | 10.1 | 17.5 | **11.6** |
| 3 | DenseCRF (Krähenbühl & Koltun, 2011) | 61 | 18.2 | 4.7 | 3.5 | 9.5 | 20.7 | 12.0 |
| | CRM (Shen et al., 2022) | 71 | 18.7 | 2.7 | 4.7 | 13.5 | 20.2 | 14.1 |
| | CascadePSP (Cheng et al., 2020) | 101 | 15.2 | 4.7 | 6.0 | 15.8 | 22.6 | **16.4** |

Table 3: **Impact of the dynamic threshold in self-correction.** If the vanilla teacher-student learning is employed in the self-correction phase (row 2), the imbalance between small and large entity segmentation is intensified which leads to worse overall performance. Our dynamic threshold for filtering the teacher's pseudo-labels (row 3) can encourage the student's predictions for small and medium entities and improve the overall AR.

| Training | AP | $AR_S$ | $AR_M$ | $AR_L$ | AR |
|---|---|---|---|---|---|
| Phase 2 | 12.8 | 8.0 | 33.7 | 43.0 | 32.6 |
| Phase 2 + 3 w/o dynamic threshold | 10.9 | 6.3 | 32.5 | **45.3** | 32.5 |
| Phase 2 + 3 w/ dynamic threshold | **12.9** | **8.6** | **35.2** | 42.0 | **33.3** |

and remove duplicates, which provides the best AR and only slightly increases the number of masks compared with the largest $\theta_{\text{merge}}$. In the second local re-clustering step, we significantly improve the recall for small entities, relatively by 168%. This step ensures that our model receives adequate supervision from small entities. We also improve the overall recall by 2.5 AR. Finally, in the third refinement step, we adopt CascadePSP (Cheng et al., 2020) because it can best boost the overall mask quality. The other two options, DenseCRF (Krähenbühl & Koltun, 2011) and CRM (Shen et al., 2022), are also viable, but they change the pseudo-labels more aggressively, leading to the removal of many potential entities. Overall, each step in our self-exploration phase contributes to the high-quality initial pseudo-labels for SOHES. Notably, we can parallelize the processing for each image and accelerate self-exploration with more compute nodes.

**Self-correction.** In our self-correction phase, we adopt a teacher-student mutual-learning framework from semi-supervised learning, to continuously improve the segmentation model by itself. However, as shown in Table 3, the initial attempt of the mutual-learning with a fixed confidence threshold leads to worse performance. In fact, the imbalanced distribution is reinforced during this procedure, as indicated by the decreased AR for small and medium entities and increased AR for larger entities. Therefore, we need a dynamic threshold that allows more small and medium pseudo-labels from the teacher model, and balances the student's prediction for entities of different scales. With the dynamic threshold, we can improve $AR_S$ and $AR_M$ relatively by 7.5% and 4.5%, with an acceptable cost of 2.3% $AR_L$. Consequently, the overall AR is improved by 0.7.

## 5 CONCLUSION

We present SOHES, a self-supervised approach towards open-world entity segmentation with hierarchical structures. Through three phases of self-evolution, a self-supervised learner is adapted to an open-world segmentation model. By recognizing and localizing entities and their constituent parts in an open world with superior mask quality, SOHES substantially closes the gap between self-supervised and supervised methods, and sets the new state of the art on various datasets.

**Acknowledgement.** This work was supported in part by NSF Grant 2106825, NIFA Award 2020-67021-32799, and the Jump ARCHES endowment through the Health Care Engineering Systems Center. This work used NVIDIA GPUs at NCSA Delta through allocations CIS220014, CIS230012, and CIS230013 from the Advanced Cyberinfrastructure Coordination Ecosystem: Services & Support (ACCESS) program, which is supported by NSF Grants #2138259, #2138286, #2138307, #2137603, and #2138296.

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

# A    IMPLEMENTATION DETAILS

**Self-exploration.** We use DINO (Caron et al., 2021) pre-trained ViT-B/8 as the feature extractor to generate patch-level visual features. During the global clustering step, we first resize unlabeled images to resolution $S \times S = 1,024 \times 1,024$, and cluster $8 \times 8$ patches with merging thresholds $\{\theta_{\text{merge},t}\}_{t=1}^{m} = \{0.6, 0.5, 0.4, 0.3, 0.2, 0.1\}$, which are decided based on the number of pseudo-labels (see Figure 8). Then in the local re-clustering step, we further investigate regions smaller than $\theta_{\text{small}}\% = 1/1,024$ of the total image area, and crop local images around them. The local image is resized to $S' \times S' = 256 \times 256$ and its subregions are clustered. In the next step, we use the off-the-shelf CascadePSP (Cheng et al., 2020) model to refine the masks. In the final step of hierarchy analysis, the coverage threshold is set to $\theta_{\text{cover}}\% = 90\%$.

**Self-instruction.** We learn a segmentation model composed of DINO (Caron et al., 2021) pre-trained ViT-B/8, ViT-Adapter (Chen et al., 2022), Mask2Former (Cheng et al., 2022), and our ancestor prediction head. The model is trained on 8 compute nodes, each equipped with 8 NVIDIA A100 GPUs. The total batch size is 128, and the number of training steps is 40,000. We optimize the model with the Adan optimizer (Xie et al., 2022) and a base learning rate of 0.0008. The total training time is about 3 days.

**Self-correction.** The teacher-student mutual learning starts after the self-instruction phase. It trains the model for additional 5,000 iterations. The teacher is updated as the exponential moving average of the student, with momentum $m = 0.9995$. The loss terms from the initial pseudo-labels and the teacher's pseudo-labels are weighted equally. In the dynamic threshold, we set $\theta_{\text{score, large}} = 0.7, \theta_{\text{score, small}} = 0.3, \gamma = 200$.

# B    DEFICIENCY OF AP METRIC IN OPEN-WORLD SEGMENTATION

The MS-COCO (Lin et al., 2014) style average precision (AP) is a prevalent metric for evaluating object detection and instance segmentation models in the traditional *closed-world* setting with a pre-defined scope of categories. However, for the *open-world* segmentation task, the AP metric becomes misleading and cannot accurately reflect the open-world model's true performance: When evaluating open-world segmentation models (which try to segment "everything") on datasets with closed-world annotations (such as MS-COCO/LVIS which only include a pre-defined, limited set of entity classes), AP would *penalize* model predictions that are actually *valid entities*, but just not annotated by the dataset.

Consequently, when the dataset annotations cannot cover all entities, AP becomes misleading for judging the performance of open-world models. As an example, the AP of SAM (Kirillov et al., 2023) is lower than CutLER (Wang et al., 2023) on MS-COCO (6.1 *vs.* 9.8, Table 4) and PartImageNet (3.4 *vs.* 5.3, Table 5), which definitely cannot imply SAM is a model inferior to CutLER. The lower AP of SAM is due to insufficient annotations in these two datasets, not the capability of SAM. Meanwhile, on datasets which explicitly try to mitigate these annotation limitations like SA-1B (Kirillov et al., 2023), AP can be more reliable than on traditional closed-world datasets like MS-COCO (Lin et al., 2014). For instance, AP on SA-1B (the most densely annotated dataset evaluated in our work, see Table 4) matches qualitative comparison and follows the same trend as AR, and our AP (12.9) significantly surpasses CutLER's AP (7.8). Prior work (Cao et al., 2023) has made a similar observation about the deficiency of MS-COCO AP evaluation in the context of self-supervised object detection.

In general, we choose the average recall (AR) as the main metric instead of AP, because AR does not penalize valid open-world predictions. Note that this choice of AR over AP is also commonly adopted in the open-world literature. Examples include (but are not limited to) detection (Kim et al., 2022), segmentation (Wang et al., 2022a), and tracking (Liu et al., 2022).

# C    ADDITIONAL EVALUATION RESULTS

We provide additional evaluation results with more metrics in Table 4 and Table 5, as a supplement to Table 1 in the main paper.

Table 4: Detailed zero-shot evaluation results on image datasets with annotations of **whole entities**.

| Method | | Mask Quality | | | |
| --- | --- | --- | --- | --- | --- |
| | AP | $AR_S$ | $AR_M$ | $AR_L$ | AR |
| *MS-COCO* (Lin et al., 2014) | | | | | |
| SAM (Kirillov et al., 2023) | 6.1 | 33.4 | 59.6 | 64.1 | 49.6 |
| FreeSOLO (Wang et al., 2022b) | 4.3 | 0.5 | 11.5 | 31.2 | 11.6 |
| CutLER (Wang et al., 2023) | **9.8** | 13.1 | 31.6 | **49.3** | 28.1 |
| HASSOD (Cao et al., 2023) | 6.0 | 14.0 | **34.1** | 45.2 | 28.3 |
| SOHES (Ours) | 2.1 | **19.8** | 31.0 | 48.8 | **30.5** |
| *LVIS* (Gupta et al., 2019) | | | | | |
| SAM (Kirillov et al., 2023) | 6.7 | 31.1 | 71.3 | 74.6 | 46.1 |
| FreeSOLO (Wang et al., 2022b) | 1.9 | 0.2 | 9.2 | 31.7 | 5.9 |
| CutLER (Wang et al., 2023) | 3.6 | 11.3 | 31.1 | 46.2 | 20.2 |
| HASSOD (Cao et al., 2023) | **4.2** | 12.7 | 36.1 | 47.8 | 22.5 |
| SOHES (Ours) | 1.9 | **19.8** | **39.4** | **59.2** | **29.1** |
| *ADE20K* (Zhou et al., 2017) | | | | | |
| SAM (Kirillov et al., 2023) | 7.8 | 31.6 | 59.2 | 62.5 | 45.8 |
| FreeSOLO (Wang et al., 2022b) | 2.3 | 0.5 | 9.3 | 27.1 | 7.3 |
| CutLER (Wang et al., 2023) | 5.2 | 15.3 | 34.7 | 44.5 | 26.3 |
| HASSOD (Cao et al., 2023) | **7.0** | 16.2 | 36.7 | 46.7 | 27.8 |
| SOHES (Ours) | 2.6 | **21.8** | **37.2** | **49.0** | **31.1** |
| *EntitySeg* (Qi et al., 2023) | | | | | |
| SAM (Kirillov et al., 2023) | 14.8 | 11.0 | 25.0 | 55.6 | 45.9 |
| FreeSOLO (Wang et al., 2022b) | 3.0 | 0.1 | 1.2 | 10.7 | 8.0 |
| CutLER (Wang et al., 2023) | **7.7** | 6.1 | 15.3 | 27.2 | 23.1 |
| HASSOD (Cao et al., 2023) | 6.1 | 7.6 | 20.1 | 30.2 | 26.2 |
| SOHES (Ours) | 5.0 | **9.1** | **21.6** | **39.6** | **33.5** |
| *SA-1B* (Kirillov et al., 2023) | | | | | |
| SAM (Kirillov et al., 2023) | 38.9 | 20.0 | 59.9 | 82.2 | 60.8 |
| FreeSOLO (Wang et al., 2022b) | 1.5 | 0.0 | 0.2 | 6.9 | 2.2 |
| CutLER (Wang et al., 2023) | 7.8 | 4.9 | 13.9 | 28.5 | 17.0 |
| HASSOD (Cao et al., 2023) | **13.8** | **12.9** | 22.8 | 38.3 | 26.0 |
| SOHES (Ours) | 12.9 | 8.6 | **35.2** | **42.0** | **33.3** |

Table 5: Detailed zero-shot evaluation results on image datasets with annotations of **object parts**.

| Method | | Mask Quality | | | |
| --- | --- | --- | --- | --- | --- |
| | AP | $AR_S$ | $AR_M$ | $AR_L$ | AR |
| *PartImageNet* (He et al., 2022a) | | | | | |
| SAM (Kirillov et al., 2023) | 3.4 | 25.4 | 29.3 | 28.5 | 28.3 |
| FreeSOLO (Wang et al., 2022b) | 3.3 | 0.6 | 7.7 | 26.4 | 13.8 |
| CutLER (Wang et al., 2023) | **5.3** | 13.2 | 26.5 | 38.1 | 28.7 |
| HASSOD (Cao et al., 2023) | 4.5 | 19.0 | **32.9** | 38.7 | 32.6 |
| SOHES (Ours) | 1.2 | **30.3** | 32.4 | **42.3** | **36.0** |
| *PACO-LVIS* (Ramanathan et al., 2023) | | | | | |
| SAM (Kirillov et al., 2023) | 1.0 | 11.9 | 34.6 | 41.1 | 18.1 |
| FreeSOLO (Wang et al., 2022b) | 0.2 | 0.1 | 5.8 | 21.3 | 2.4 |
| CutLER (Wang et al., 2023) | 0.2 | 5.0 | 18.7 | 25.1 | 8.9 |
| HASSOD (Cao et al., 2023) | **0.4** | 6.4 | 24.2 | 31.4 | 11.4 |
| SOHES (Ours) | **0.4** | **12.0** | 29.0 | 41.9 | **17.1** |

# D   ADDITIONAL ABLATION STUDY RESULTS

Since SOHES is a self-supervised approach, we do *not* base our selection of hyper-parameters on *a posteriori* model performance. Instead, we choose hyper-parameters by considering simple criteria such as computation constraints or small-scale experiments. In this section, we detail the design choices in SOHES.

**Merging thresholds.** In the first step of our self-exploration phase, we cluster patches into coherent regions and stop at pre-set merging thresholds $\theta_{\text{merge},1} > \cdots > \theta_{\text{merge},m}$. We choose these thresholds based on a practical computation constraint: pseudo-labels per image. When there are too many pseudo-labels, data loading, augmentation, and pre-processing would become a bottleneck in model training. Therefore, we control the number of pseudo-labels per image to be under 200. As shown in Figure 8, the quantity of pseudo-labels grows significantly when the merging threshold is larger. In order not to exceed 200 masks per image, we set the merging thresholds as $0.6, 0.5, 0.4, 0.3, 0.2, 0.1$.

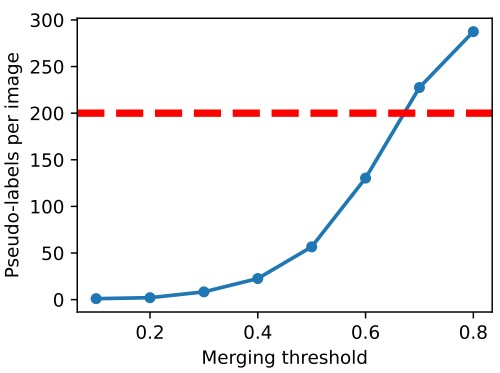

Figure 8: **Relation between the number of pseudo-labels and the merging thresholds.** As the merging threshold increases, the number of pseudo-labels rapidly grows. To control the number of pseudo-labels per image, we choose merging thresholds no larger than 0.6.

**Local re-clustering threshold.** In the second step of self-exploration, we use a threshold $\theta_{\text{small}}\%$ to select small regions that require a local re-examination. Pseudo-labels with areas smaller than $\theta_{\text{small}}\%$ of the whole image are processed via local re-clustering, and this step improves the coverage of small entities. We choose $\theta_{\text{small}}\% = 1/1,024$ because 1) the commonly adopted MS-COCO style evaluation (Lin et al., 2014) defines small objects as objects whose area is smaller than $32\times32$ pixels; 2) our images are resized to $1,024 \times 1,024$ in the self-exploration phase; 3) $(32 \times 32)/(1,024 \times 1,024) = 1/1,024$. In fact, if the threshold $\theta_{\text{small}}\%$ is larger, the coverage of small entities could not be significantly improved further, and the processing time would be longer, as shown in Table 6a.

Table 6: **Impact of hyper-parameters $\theta_{\text{small}}\%$ and $\theta_{\text{cover}}\%$.**

(a) **Choice of the local re-clustering threshold $\theta_{\text{small}}\%$.** We set $\theta_{\text{small}}\% = 1/1,024$ by considering the relative areas of small entities. Using a larger $\theta_{\text{small}}\%$ does not further improve the coverage of small entities, but introduces additional computation overheads.

| $\theta_{\text{small}}\%$ | $\text{AR}_S$ | Time/Img (sec) |
|---|---|---|
| 0 (No local re-clustering) | 1.9 | 0.0 |
| $1/2,048$ | 4.3 | 6.0 |
| $1/1,024$ | **5.1** | 8.4 |
| $1/512$ | **5.1** | 10.0 |

(b) **Choice of the coverage threshold $\theta_{\text{cover}}\%$.** We set $\theta_{\text{cover}}\% = 0.9$ for robust hierarchical relations between our pseudo-labels. When $\theta_{\text{cover}}\%$ varies within $[0.6, 0.95]$, the hierarchical relations are stable.

| $\theta_{\text{cover}}\%$ | Relative Change w.r.t. $\theta_{\text{cover}} = 0.9$ |
|---|---|
| 0.6 | 9% |
| 0.7 | 7% |
| 0.8 | 4% |
| 0.85 | 2% |
| 0.9 | 0% |
| 0.95 | 4% |
| 1.0 | 36% |

**Coverage threshold.** In the final step of self-exploration, we analyze the hierarchical relations between pseudo-labels. The pairwise test involves a coverage threshold $\theta_{\text{cover}}\%$. For robustness, we

choose $\theta_{\text{cover}}\% = 0.9$ to allow an ancestor pseudo-label to not necessarily cover all of its descendants' pixels. Indeed, when $\theta_{\text{cover}}\% \in [0.6, 0.95]$, the induced hierarchical relations are relatively stable, as shown in Table 6b.

**DINO backbone.** As a supplement to Figure 7 in the main paper, Table 7 summarizes the detailed statistics of the mask quality of initial pseudo-labels produced by DINO backbones pre-trained with different configurations. It also includes the mask quality after each self-exploration step.

Table 7: **Comparison of DINO backbones pre-trained with different configurations.** A small pre-training patch size is critical for producing fine-grained visual features and high-quality pseudo-label masks. The steps 1, 2, and 3 refer to global clustering, local re-clustering, and mask refinement, respectively.

| Pre-training | | Step | Masks/Image | Mask Quality | | | | |
| Resolution | Patch Size | | | AP | $AR_S$ | $AR_M$ | $AR_L$ | AR |
|---|---|---|---|---|---|---|---|---|
| 224 | 8 | 1 | 144 | 0.8 | 1.6 | 5.3 | 14.7 | **7.6** |
| | | 2 | 124 | 1.3 | 4.1 | 7.8 | 15.0 | **9.4** |
| | | 3 | 105 | 3.2 | 5.0 | 12.6 | 20.6 | **13.8** |
| 224 | 16 | 1 | 108 | 0.8 | 1.3 | 4.5 | 13.0 | 6.6 |
| | | 2 | 75 | 1.7 | 3.5 | 6.9 | 13.3 | 8.3 |
| | | 3 | 65 | 4.0 | 4.2 | 11.3 | 17.0 | 11.9 |
| 224 | 32 | 1 | 43 | 0.6 | 0.7 | 2.6 | 5.8 | 3.3 |
| | | 2 | 17 | 0.9 | 0.3 | 2.2 | 5.9 | 3.0 |
| | | 3 | 14 | 1.8 | 0.4 | 3.4 | 6.6 | 3.9 |
| 448 | 8 | 1 | 114 | 1.0 | 1.2 | 4.2 | 14.6 | 6.9 |
| | | 2 | 103 | 1.6 | 3.1 | 6.4 | 14.8 | 8.5 |
| | | 3 | 82 | 3.4 | 3.9 | 10.8 | 19.1 | 12.2 |
| 448 | 16 | 1 | 102 | 0.8 | 1.3 | 3.9 | 13.1 | 6.3 |
| | | 2 | 84 | 1.5 | 3.8 | 6.7 | 13.4 | 8.3 |
| | | 3 | 71 | 3.8 | 4.6 | 10.8 | 17.1 | 11.8 |
| 448 | 32 | 1 | 48 | 0.7 | 0.8 | 2.2 | 7.1 | 3.5 |
| | | 2 | 25 | 1.0 | 1.1 | 2.5 | 7.2 | 3.7 |
| | | 3 | 20 | 2.4 | 1.4 | 4.1 | 8.3 | 4.9 |

**Segmentation head.** In SOHES, we choose Mask2Former (Cheng et al., 2022) as our segmentation head mainly for making hierarchical predictions. In Cascade Mask R-CNN (Cai & Vasconcelos, 2018) used by prior work (Wang et al., 2023; Cao et al., 2023), each proposal is predicted independently, and therefore, analyzing hierarchical relations between entities would be challenging. In contrast, the attention modules in Mask2Former allow information exchange among queries, so we can build our ancestor prediction head on Mask2Former (see Figure 4). For a more comprehensive comparison with prior work, we train a segmentation model based on Cascade Mask R-CNN using SOHES without the entity hierarchy. As shown in Table 8, this model still significantly outperforms CutLER and HASSOD with the same segmentation head.

When using the Mask2Former segmentation head, we can extend it with our proposed ancestor prediction head to perform the additional task of hierarchical relation predictions. This module can be considered as an add-on to the original segmentation head with minimal influence on the mask quality, since this ancestor prediction head operates in parallel to the mask prediction head. In fact, the segmentation performance of a SOHES model trained *without* the ancestor prediction head is close to that of the standard SOHES model (*e.g.*, on SA-1B, 33.0 Mask AR without ancestor prediction *vs.* 33.3 Mask AR with ancestor prediction), but the former model cannot predict hierarchical relations.

Table 8: **Comparison of models trained with the Cascade Mask R-CNN segmentation head.** In the main paper, we choose the Mask2Former segmentation head to model hierarchical relations between predictions, while prior methods usually use Cascade Mask R-CNN. With the same Cascade Mask R-CNN segmentation head, SOHES still outperforms prior work.

| Method | Segmentation Head | Mask Quality (AR) | | |
|---|---|---|---|---|
| | | LVIS | EntitySeg | SA-1B |
| CutLER (Wang et al., 2023) | Cascade Mask R-CNN | 20.2 | 23.1 | 17.0 |
| HASSOD (Cao et al., 2023) | (Cai & Vasconcelos, 2018) | 22.5 | 26.2 | 26.0 |
| SOHES (Ours) | | **27.0** | **29.2** | **31.9** |
| SOHES (Ours) | Mask2Former (Cheng et al., 2022) | **29.1** | **33.5** | **33.3** |

## E LIMITATIONS

In Figure 9, we visualize some failure cases of SOHES: 1) When there are discontinuous entities or occlusion (*e.g.*, sky separated by foreground objects), SOHES may not correctly associate the disconnected segments of the same entity. The reason is that in the self-exploration phase, we only merge adjacent regions. The model rarely observes one entity separated in multiple disconnected regions. We observe that the copy-paste data augmentation (Ghiasi et al., 2021) can simulate occlusion and partially mitigate this issue, so we have adopted such data augmentation in SOHES training. 2) SOHES often produces imprecise segmentation masks for letters and characters. The boundary is not perfectly aligned with strokes and the mask is often larger than the letter. 3) When there are blurry backgrounds, SOHES tends not to predict a mask for the background. Similar failure cases can be observed when applying prior methods (Wang et al., 2023; Cao et al., 2023; Kirillov et al., 2023) to images affected by occlusion, text overlays, or blurring. We aim to resolve these issues with improved pseudo-labeling strategies in future work.

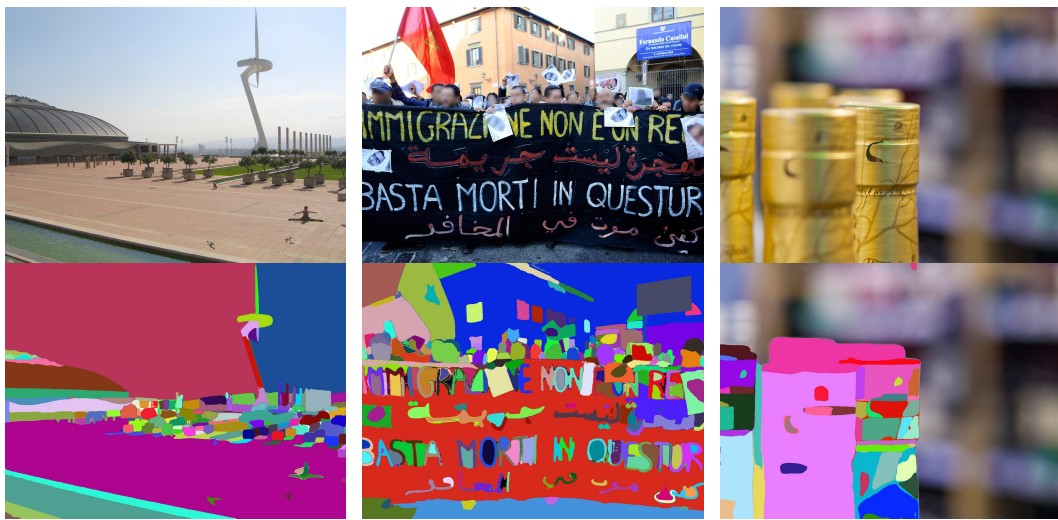

Figure 9: **Limitations of SOHES.** Our method sometimes fail to accurately segment discontinuous or occluded entities (left), letters or characters (middle), and blurred background (right).

## F QUALITATIVE RESULTS

In Figures 10, 11, 12, 13, and 14, we visualize the segmentation results of SOHES, and qualitatively compare SOHES with the supervised model SAM on the evaluation datasets we have used in the main paper.

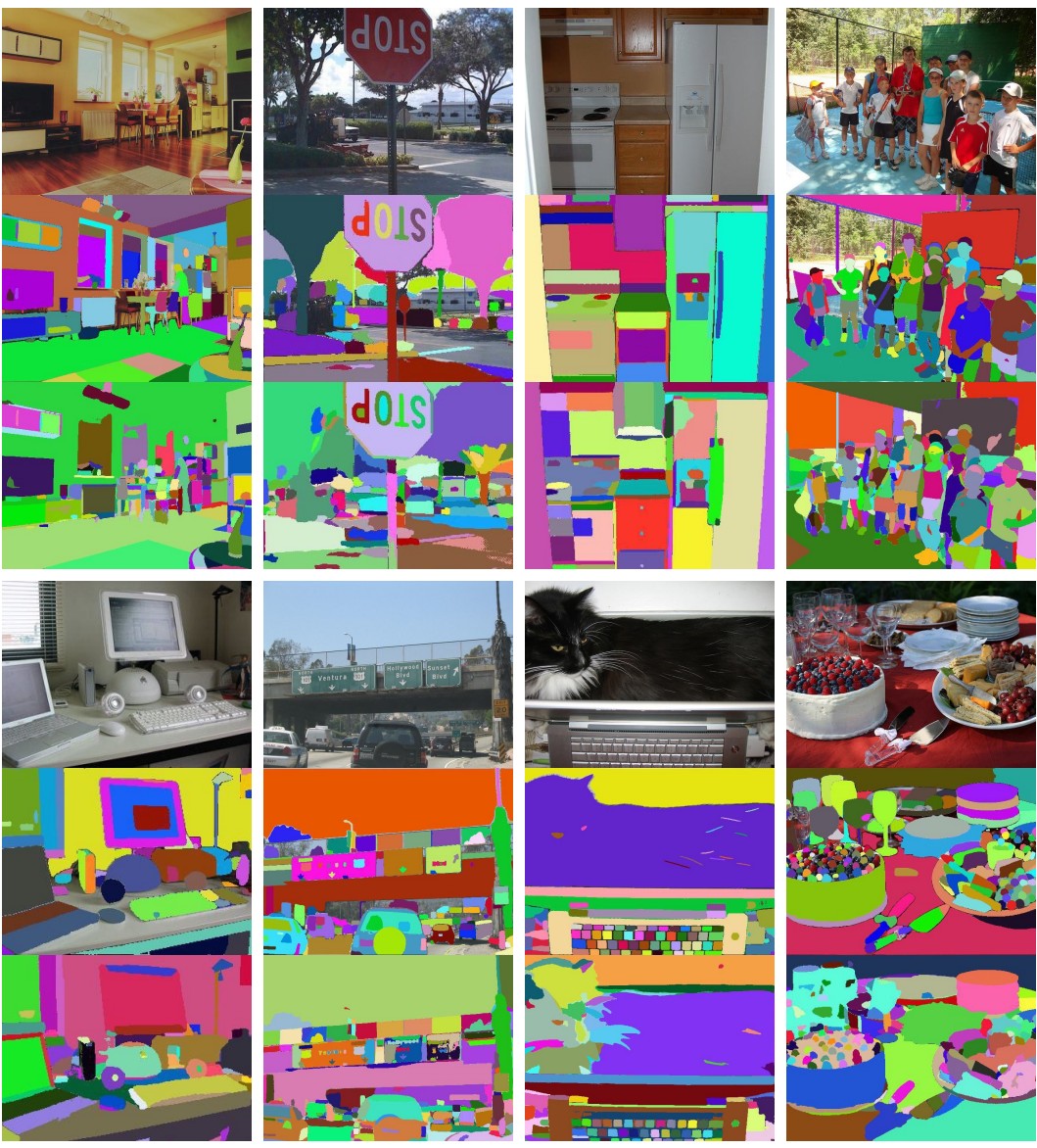

Figure 10: **Qualitative results on MS-COCO images.** In each group of images, from top to bottom we show the original input image, segmentation by SAM (Kirillov et al., 2023), and segmentation by SOHES. As a self-supervised method, SOHES can achieve results that are comparable to those produced by the supervised model SAM.

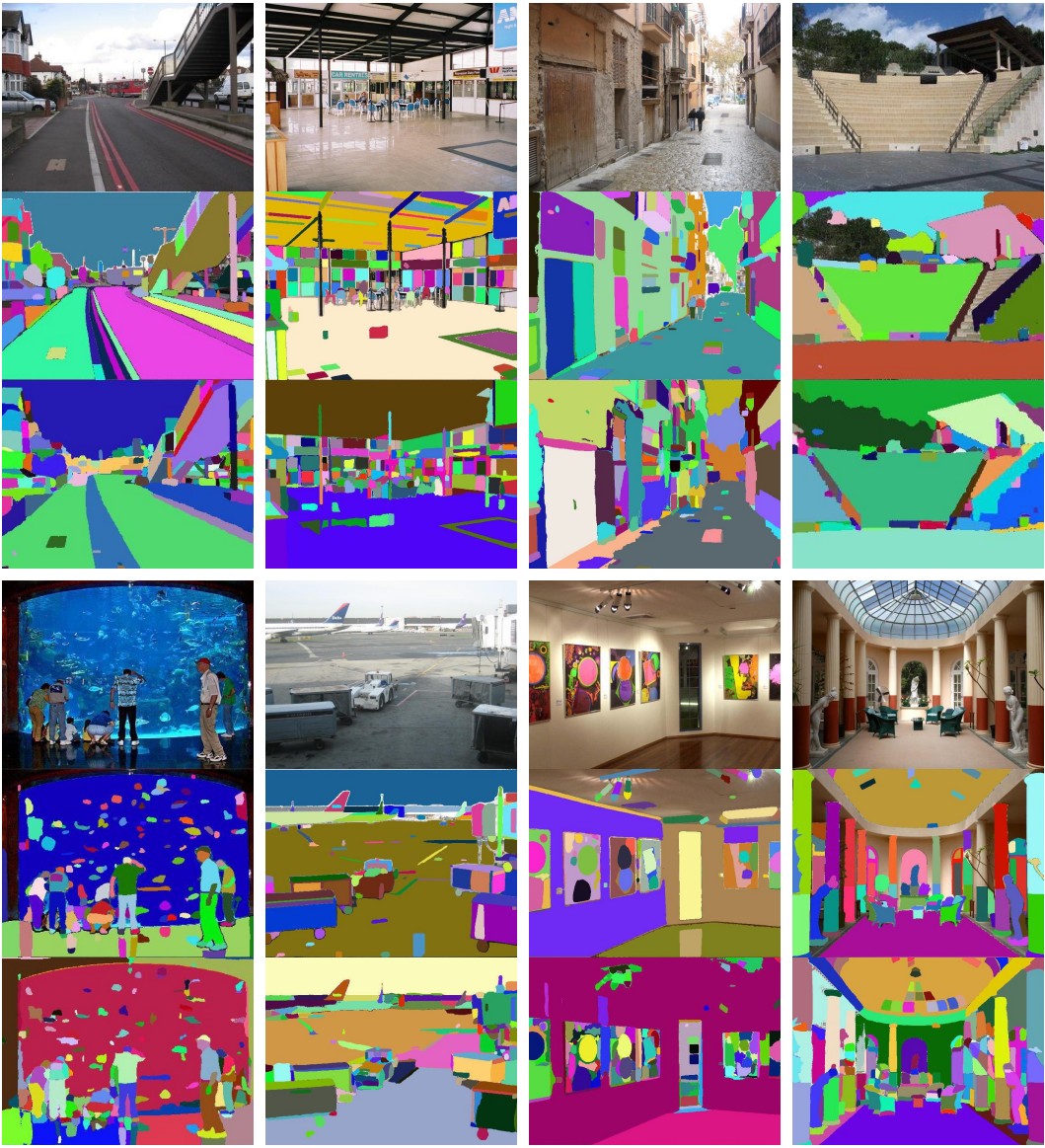

Figure 11: **Qualitative results on ADE20K images.** In each group of images, from top to bottom we show the original input image, segmentation by SAM (Kirillov et al., 2023), and segmentation by SOHES. As a self-supervised method, SOHES can achieve results that are comparable to those produced by the supervised model SAM.

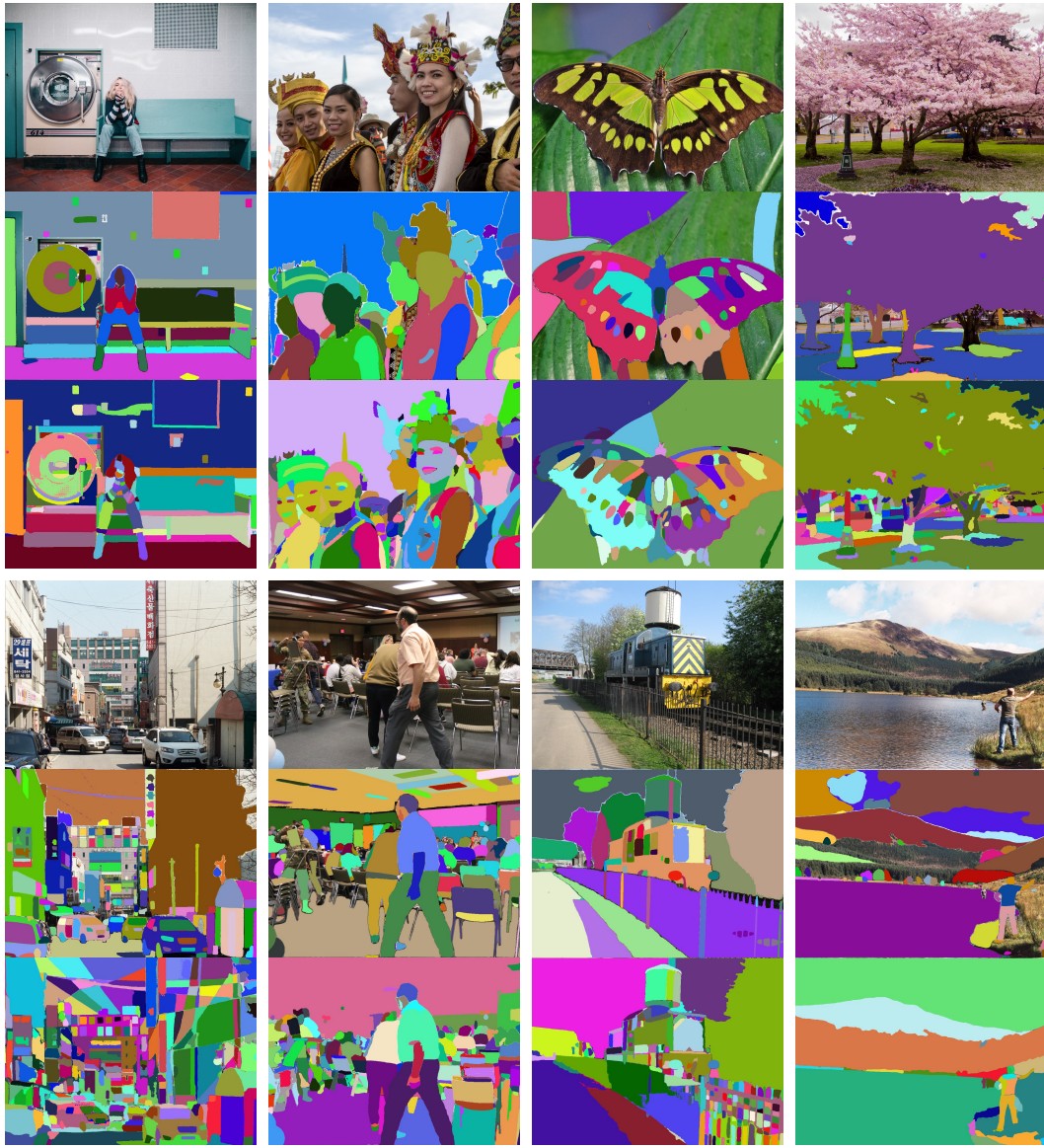

Figure 12: **Qualitative results on EntitySeg images.** In each group of images, from top to bottom we show the original input image, segmentation by SAM (Kirillov et al., 2023), and segmentation by SOHES. As a self-supervised method, SOHES can achieve results that are comparable to those produced by the supervised model SAM.

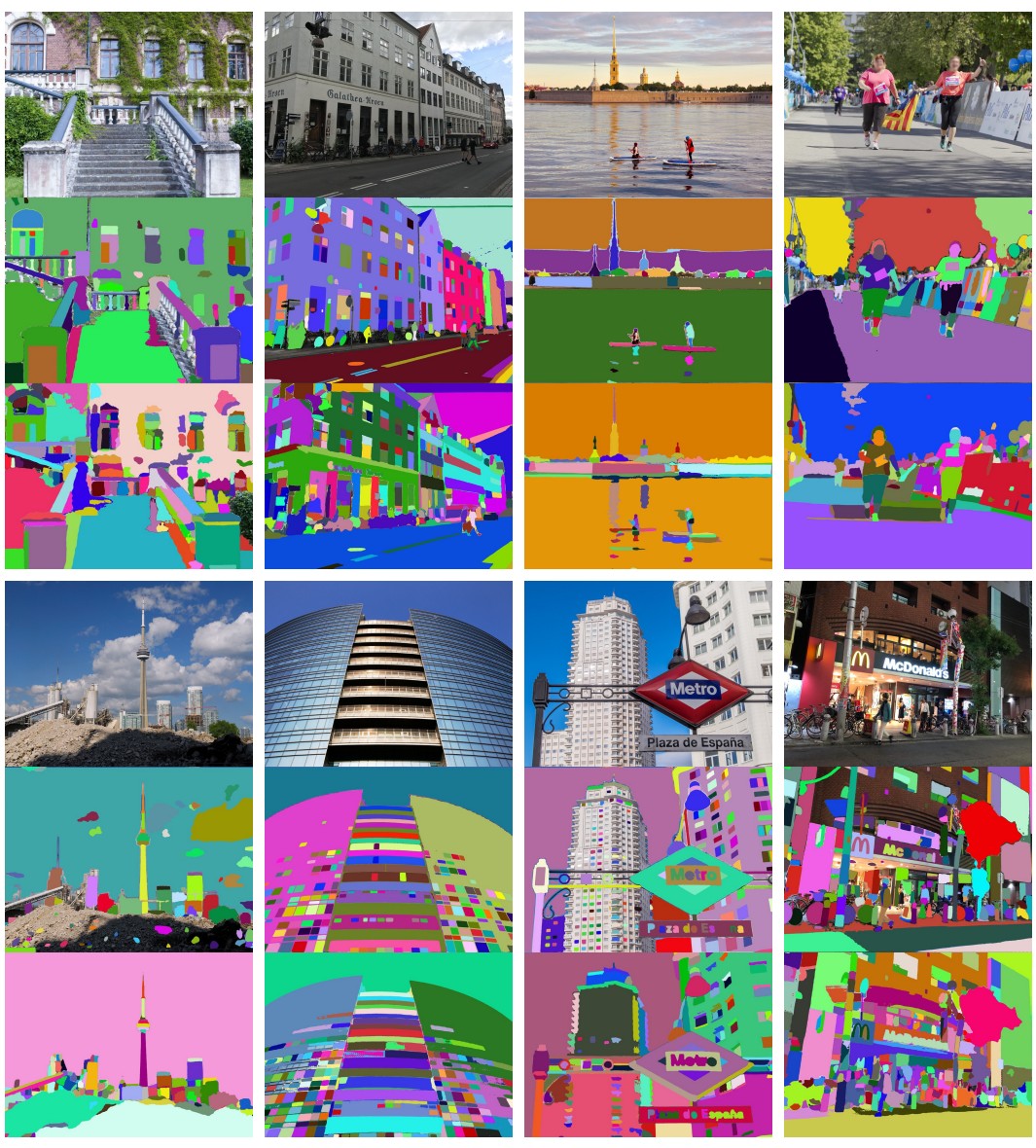

Figure 13: **Qualitative results on SA-1B images.** In each group of images, from top to bottom we show the original input image, segmentation by SAM (Kirillov et al., 2023), and segmentation by SOHES. As a self-supervised method, SOHES can achieve results that are comparable to those produced by the supervised model SAM.

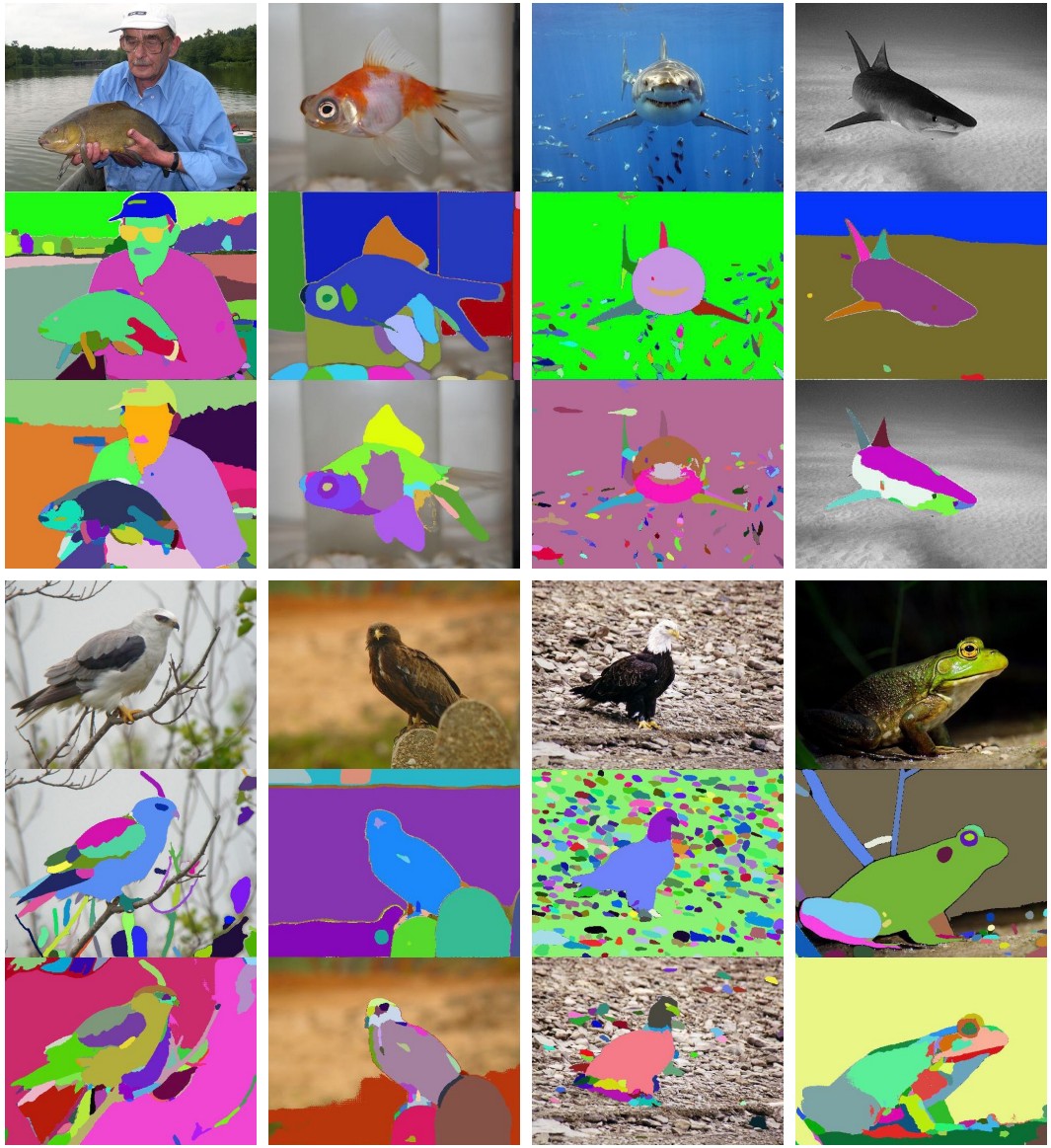

Figure 14: **Qualitative results on PartImageNet images.** In each group of images, from top to bottom we show the original input image, segmentation by SAM (Kirillov et al., 2023), and segmentation by SOHES. As a self-supervised method, SOHES can achieve results that are comparable to those produced by the supervised model SAM.

