# OpenReview forum: "SOHES: Self-supervised Open-world Hierarchical Entity Segmentation"
_ICLR.cc/2024/Conference — ICLR 2024 poster_

### Official Review · Reviewer_Rono · 2023-10-29

**Soundness:** 3 good
**Presentation:** 3 good
**Contribution:** 2 fair
**Rating:** 6
**Confidence:** 3

**Summary:**

The paper introduces SOHES, a method designed to eliminate the need for manual annotation and facilitate the learning of hierarchical associations for open-world entity segmentation. Specifically, the authors have devised a novel approach for generating initial pseudo-labels and subsequently enhancing the segmentation model through their utilization. When compared to previous research on open-world entity segmentation, SOHES demonstrates a significant reduction in the gap between self-supervised methods and the supervised SAM.

**Strengths:**

1. The proposed method is highly motivated, and the results of SOHES demonstrate tremendous potential in open-world entity segmentation, even surpassing SAM's performance on certain datasets.
2. The paper is excellently structured and provides a clear and easy-to-follow presentation.

**Weaknesses:**

1. The analysis of the hierarchical architecture is inadequate.
2. Certain details in the method lack clarity, and there is a noticeable absence of some ablation experiments.

**Questions:**

1. Regarding the generation of pseudo-labels, I am inquisitive about why the visual feature of merged patches is computed as the sum of original features rather than an average or any other operation?
2. Concerning the second local re-clustering step, I'm curious about whether it has an impact on the results for large entities and how the threshold for small regions affects the final results.
4. In the ablation study, there is no information regarding how the ancestor prediction head contributes to the final results. Further analysis of the hierarchical architecture is needed for clarification.

---

> ### Author Response · Authors · 2023-11-19
> **Response to Reviewer Rono**
>
> We appreciate the detailed feedback you provided for our submission. We are encouraged by your acknowledgement of our motivation, experimental results, and overall presentation. We provide the following clarifications in response to your concerns:
>
> 1. Sum of features
>
> - In fact, summing and averaging the features when merging two regions are equivalent. The reason is that, we adopt the *cosine similarity* when measuring the semantic closeness between two regions, which normalizes the features’ magnitude in computation. The sum of patch-level features within a region leads to the same cosine similarity as the average of patch-level features.
>
> - We chose the sum over average due to its calculation and implementation simplicity. Equivalent to computing and storing the feature sum $f_k=f_i+f_j$, we can also use the feature average $\bar{f_k}=\frac{N_i\bar{f_i}+N_j\bar{f_j}}{N_i+N_j}$, where $N_i,N_j$ are the numbers of patches in region $i$ and $j$, respectively. The sum formation does not require keeping track of the patch numbers.
>
> 2. Impact of local re-clustering
>
> - We designed the local re-clustering to refine the pseudo-labels for small entities, because they are the most challenging to be discovered. The local re-clustering has minimal impact on the large entities. As shown in Table 2 of the original manuscript, the average recall for large entities (AR-Large) is slightly improved from 17.2 to 17.5 by the local re-clustering step. The reason for this change is that “large” entities considered by the MS-COCO evaluation metric are regions that are larger than $96\times 96$ pixels (https://cocodataset.org/#detection-eval), which could still fall under our relative area threshold $\theta_\text{small}=\frac{1}{1024}$ in some high-resolution images from SA-1B (e.g., $1500\times 8000$). Visually, these entities are not “large” compared to the whole image, but are still counted by MS-COCO style AR$_L$.
>
> - To better understand the impacts of the hyper-parameter $\theta_\text{small}$, we run additional ablation study experiments. We observe that a threshold larger than $\frac{1}{1024}$ does not further significantly improve the coverage of small entities but would introduce more processing time.
> | $\theta_\text{small}$ | 0 (No local re-clustering) | $\frac{1}{2048}$ | $\frac{1}{1024}$ | $\frac{1}{512}$ |
> | --------------------- | -------------------------- | ---------------- | ---------------- | --------------- |
> | AR$_S$            	| 1.9                    	| 4.3          	| 5.1          	| 5.1         	|
> | Time/Image (sec)  	| 0.0                      	| 6.0          	| 8.4          	| 10.0        	|
>
> - We appreciate the reviewer’s feedback and will integrate the explanation for the impact of $\theta_\text{small}$ into the revision.
>
> 3. Ablating the ancestor prediction head
>
> - We proposed the ancestor prediction head to perform the additional task of hierarchical segmentation, which could be considered as an add-on extending the Mask2Former segmentation model. We did not expect it to help with mask prediction, since this ancestor prediction head operates in parallel to the mask prediction head. In fact, we trained a SOHES model without the ancestor prediction head, and its performance is very close to the standard SOHES model (e.g., on SA-1B, 33.0 Mask AR w/o ancestor prediction vs. 33.3 Mask AR w/ ancestor prediction).

---

> > ### Comment · Reviewer_Rono · 2023-11-21
> >
> > Thanks for the replies, after reading others' comments and feedback, I have updated my rating.

---

> > > ### Author Response · Authors · 2023-11-21
> > > **Thank you!**
> > >
> > > Thank you very much for reviewing our discussions and updating the rating. We are glad that our response has addressed your concerns and appreciate your recognition of our paper.

---

### Official Review · Reviewer_vWUT · 2023-10-31

**Soundness:** 3 good
**Presentation:** 3 good
**Contribution:** 3 good
**Rating:** 6
**Confidence:** 3

**Summary:**

The paper introduces the Self-supervised Open-world Hierarchical Entity Segmentation (SOHES) approach for computer vision, which can segment entities in images beyond pre-defined classes without human annotations. SOHES uses three stages: self-exploration, self-instruction, and self-correction, generating high-quality pseudo-labels from visual feature clustering and refining them through mutual-learning. This method achieves a new standard in self-supervised open-world segmentation, eliminating the need for human-annotated masks.

**Strengths:**

1.	The task SOHES is seldom investigated.
2.	This paper proposed a new method to generate hierarchical proposals.
3.	This paper proposed an ancestor prediction head, which is novel.
4.	The proposed method significantly outperformed the previous methods.

**Weaknesses:**

1.	Although this paper divide the stages into self-exploration, self-instruction, and self-correction. But it looks like previous papers[1] that generate pseudo-labels, then training from pseudo-labels, and apply self-training to improve the model. The framework is actually quite common. So what is the core difference from the previous works in the framework?
2.     The authors claimed that “Existing segmentation models cannot predict the hierarchical relations among masks. ” However, methods like Groupvit[2] already can predict the hierarchical relations among masks.
3.	The proposed method introduced too many hyperparameters such as Theta_{merge}, Theta_{small}, Theta_{cover}. There is not explanation for the selection of some hyperparameters, such as Theta_{cover}.
4.	Although the authors proposed the ancestor prediction head for the hierarchical segmentation. However, the mask prediction of masks at each hierarchy is independent.  I am curious if the relation modeling of hierarchy can contribute to better mask predictions. I am also curious about the effect of just predicting the hierarchical relationships. It seems that there is no ablation to verify if the ancestor prediction head can bring the performance improvements.

[1] Wang, Xinlong, et al. "Freesolo: Learning to segment objects without annotations." Proceedings of the IEEE/CVF Conference on Computer Vision and Pattern Recognition. 2022.
[2] Xu, Jiarui, et al. "Groupvit: Semantic segmentation emerges from text supervision." Proceedings of the IEEE/CVF Conference on Computer Vision and Pattern Recognition. 2022.

**Questions:**

See weakness

---

> ### Author Response · Authors · 2023-11-19
> **Response to Reviewer vWUT (Part 1)**
>
> We appreciate the detailed feedback you provided for our submission. We are encouraged by your acknowledgement of our novel task, “method to generate hierarchical proposals,” “novel ancestor prediction head,” and good performance. We provide the following clarifications in response to your concerns:
>
> 1. Multi-phase framework
>
> - We agree with the reviewer that such a multi-phase learning paradigm (e.g., discovering pseudo-labels and then learning a detection/segmentation model) is a general framework in the existing self-supervised object localization/discovery literature. Recent methods which adopt this framework include LOST [R1], TokenCut [R2], FreeSOLO [R3] (as suggested by the reviewer), FOUND [R4], and CutLER [R5] (as mentioned by Reviewer PuvF). The contributions of these different methods do not lie in the introduction of a new framework, but instead in proposing unique components to improve this established framework for self-supervised object localization/discovery.
>
> - Our SOHES similarly adopts this general multi-phase framework, but makes significant contributions through **novel designs for each phase** involved in the learning process, including a global-to-local clustering algorithm for pseudo-label generation, a hierarchical relation learning module, and a teacher-student self-correction phase. In particular, none of the previously mentioned methods [R1-5] consider hierarchical structures among objects/entities, while learning such structures is involved throughout the learning process of SOHES. Given these technical novelties and advancements, our SOHES substantially improves performance over previous methods.
>
> - We appreciate the reviewer’s feedback and will revise the Related Work section and the beginning of the Approach section, to better reflect this general framework and our novel components proposed in this work with a hierarchical perspective.
>
> 2. Hierarchical segmentation model
>
> - We would like to first clarify that in the original manuscript, we stated that “existing segmentation models cannot predict the hierarchical relations among masks” **in the context** of general segmentation model architectures like Mask2Former (Section 3.2). This drove us to design a novel module that could predict hierarchical structures among entities. Indeed, some *specialized* segmentation models (e.g., GroupViT [R6]) are capable of making hierarchical predictions, but do not directly fit into our context.
>
> - More importantly, although our SOHES and GroupViT both aim to produce hierarchical segmentation masks, there are some vital differences between the two approaches:
> 	- SOHES directly learns from unlabeled raw images, but GroupViT is weakly supervised by text.
> 	- SOHES can produce a variable number of hierarchical levels, but GroupViT is limited to pre-defined levels (e.g., 2 levels).
> 	- SOHES can produce a variable number of whole entities, parts, and subparts, but GroupViT is limited to a fixed number of pre-defined group tokens (e.g., 64 and 8 group tokens in the first and second grouping stages, respectively).
>
> - We appreciate the reviewer’s feedback and will revise this sentence in Section 3.2 to more precisely describe the context about the widely-applied segmentation architectures. In addition, we will cite GroupViT [R6] and include the discussion above in the revision.

---

> > ### Author Response · Authors · 2023-11-19
> > **Response to Reviewer vWUT (Part 2)**
> >
> > 3. Selection of hyper-parameters
> >
> > - We selected these hyper-parameters for simple reasons:
> > 	- $\theta_\text{merge}$: This hyper-parameter determines the stopping criterion in the clustering step. We chose $0.1, 0.2, 0.3, 0.4, 0.5, 0.6$ to cover as many entities as possible without exceeding a practical computation constraint. The reason for this choice was detailed in Appendix C of the original manuscript.
> > 	- $\theta_\text{small}$: This hyper-parameter is the threshold for small pseudo-labels that are processed via local re-clustering, which aims to improve the coverage of small entities. We chose $\frac{1}{1024}$ because a) the commonly adopted MS-COCO style evaluation defines small objects as objects whose area is smaller than $32\times 32$ pixels (https://cocodataset.org/#detection-eval), b) images are resized to $1024\times 1024$ in our pseudo-label generation phase, and c) $\frac{32\times 32}{1024\times 1024}=\frac{1}{1024}$.
> > 	- $\theta_\text{cover}$: This hyper-parameter controls the hierarchical relation test between a pair of masks. We chose 90% (rather than 100%) for better robustness. For example, even if one pixel of pseudo-label $i$ is not in pseudo-label $j$, it is still possible that $j$ is an ancestor of $i$ -- we just require that $j$ covers most of $i$’s pixels.
> >
> > - To better understand the impacts of hyper-parameters $\theta_\text{small}$ and $\theta_\text{cover}$ ($\theta_\text{merge}$ has been discussed in Table 2 and Appendix C of the original manuscript), we run additional ablation study experiments.
> > 	- $\theta_\text{small}$: We observe that a threshold larger than $\frac{1}{1024}$ does not further significantly improve the coverage of small entities but would introduce more processing time.
> > | $\theta_\text{small}$ | 0 (No local re-clustering) | $\frac{1}{2048}$ | $\frac{1}{1024}$ | $\frac{1}{512}$ |
> > | --------------------- | -------------------------- | ---------------- | ---------------- | --------------- |
> > | AR$_S$            	| 1.9                    	| 4.3          	| 5.1          	| 5.1         	|
> > | Time/Image (sec)  	| 0.0                      	| 6.0          	| 8.4          	| 10.0        	|
> > 	- $\theta_\text{cover}$: We are not able to quantitatively evaluate different hierarchical relations among pseudo-labels, due to the lack of ground-truth annotations. However, we can examine the robustness of hierarchical relations when we alter this hyper-parameter $\theta_\text{cover}$. We compare the relations induced by varying $\theta_\text{cover}$ vs. $\theta_\text{cover}=0.9$, and count the ratio of changed relations. Indeed, when $\theta_\text{cover} \in [0.6, 0.95]$, the relations are relatively stable.
> > | $\theta_\text{cover}$                        	| 0.6  | 0.7  | 0.8  | 0.85 | 0.9  | 0.95 | 1.0  |
> > | ------------------------------------------------ | ---- | ---- | ---- | ---- | ---- | ---- | ---- |
> > | Relative Change w.r.t. $\theta_\text{cover}=0.9$ | 9%   | 7%   | 4%   | 2%   | 0%   | 4%   | 36%  |
> >
> > - We appreciate the reviewer’s feedback and will integrate the explanation and ablation study for these choices of hyper-parameters into the revision.
> >
> > 4. Ablating the ancestor prediction head
> >
> > - We proposed the ancestor prediction head to perform the additional task of hierarchical segmentation, which could be considered as an add-on extending the Mask2Former segmentation model. We did not expect it to help with mask prediction, since this ancestor prediction head operates in parallel to the mask prediction head. In fact, we trained a SOHES model without the ancestor prediction head, and its performance is very close to the standard SOHES model (e.g., on SA-1B, 33.0 Mask AR w/o ancestor prediction vs. 33.3 Mask AR w/ ancestor prediction).
> >
> > - By “just predicting the hierarchical relationships,” we can only create a hierarchical structure of the queries in Mask2Former. However, these queries do not have any physical meaning before we use the mask prediction head to instantiate their corresponding masks in the given image. Thus, we can only predict the hierarchical relations in conjunction with the mask prediction, but not using the ancestor prediction head alone.

---

> > > ### Author Response · Authors · 2023-11-19
> > > **Reference in Response**
> > >
> > > [R1] Oriane Siméoni, Gilles Puy, Huy V. Vo, Simon Roburin, Spyros Gidaris, Andrei Bursuc, Patrick Pérez, Renaud Marlet, and Jean Ponce. Localizing Objects with Self-Supervised Transformers and no Labels. In BMVC, 2021.
> > >
> > > [R2] Yangtao Wang, Xi Shen, Shell Xu Hu, Yuan Yuan, James L. Crowley, and Dominique Vaufreydaz. TokenCut: Self-supervised Transformers for Unsupervised Object Discovery using Normalized Cut. In CVPR, 2022.
> > >
> > > [R3] Xinlong Wang, Zhiding Yu, Shalini De Mello, Jan Kautz, Anima Anandkumar, Chunhua Shen, and Jose M. Alvarez. FreeSOLO: Learning to Segment Objects without Annotations. In CVPR, 2022.
> > >
> > > [R4] Oriane Siméoni, Chloé Sekkat, Gilles Puy, Antonin Vobecky, Eloi Zablocki and Patrick Pérez. Unsupervised Object Localization: Observing the Background to Discover Objects. In CVPR, 2023.
> > >
> > > [R5] Xudong Wang, Rohit Girdhar, Stella X. Yu, and Ishan Misra. Cut and Learn for Unsupervised Object Detection and Instance Segmentation. In CVPR, 2023.
> > >
> > > [R6] Jiarui Xu, Shalini De Mello, Sifei Liu, Wonmin Byeon, Thomas Breuel, Jan Kautz, and Xiaolong Wang. GroupViT: Semantic Segmentation Emerges from Text Supervision. CVPR 2022.

---

> ### Comment · Reviewer_vWUT · 2023-11-22
>
> I have read the responses from the authors. Most of my issues have been addressed. However, the "ancestor prediction head" does not have significant improvements. I decided to keep my original rating.

---

> > ### Author Response · Authors · 2023-11-22
> > **Thank you!**
> >
> > We appreciate your time in reviewing our work and participating in the discussion. We are glad that our responses have addressed most of your issues.
> >
> > Regarding the ancestor prediction head, we would like to kindly emphasize the novelty and significance in proposing this module: Although it does not greatly increase the Mask AR score, the ancestor prediction head has accomplished its key function – to produce hierarchical relations between segmentation masks, leading to a structured analysis of entities and their constituent parts in the given image. **By proposing the ancestor prediction, we enable a brand new capability of hierarchical segmentation which did not exist in the current self-supervised object discovery literature [R1-5].**

---

### Official Review · Reviewer_PuvF · 2023-11-02

**Soundness:** 2 fair
**Presentation:** 2 fair
**Contribution:** 3 good
**Rating:** 6
**Confidence:** 5

**Summary:**

The authors introduced the Self-supervised Open-world Hierarchical Entity Segmentation (SOHES) method, a three-phase approach for entity segmentation. The first phase, Self-exploration, uses a pre-trained DINO model to produce initial pseudo-labels. By clustering visual features, it identifies regions representing meaningful entities. In the Self-instruction phase, a Mask2Former segmentation model refines the segmentation by training on these initial labels. Even with some label noise, the model effectively averages out inconsistencies, resulting in better mask predictions. The final Self-correction phase uses a teacher-student mutual-learning framework to further refine the model's predictions and adapt to open-world segmentation. This approach only uses raw images without human annotations. A standout feature of SOHES is its ability to segment not just whole entities but also their parts and sub-parts.

**Strengths:**

[Task] Unsupervised image segmentation holds significant importance, and this study successfully performs segmentation without human supervision, offering segmentation masks at multiple levels of granularity.

[The generation of hierarchical masks] The approach to generate unsupervised hierarchical masks is pretty interesting. And surprisingly, this method surpassed SAM in recall on some evaluation benchmarks.

[Paper writing] The paper is well-articulated, effectively communicating the central ideas.

**Weaknesses:**

[Technical Contributions] The three phases proposed in this work are very similar to the Cut-and-Learn pipeline proposed by CutLER [1]. Self-exploration is pretty similar to the MaskCut stage in CutLER, which also leverages DINO feature for pseudo-label generation. Self-instruction is the same as the LEARN process of CutLER, which trains a model on pseudo-labels. And, the Self-correction stage can be viewed as a variant of CutLER's multi-round self-training, but with a teacher-student framework. I agree that while there are some implementation differences between the stages in SOHES and CutLER, their core concepts are largely analogous.

[Model performance] SOHES performs much worse than CutLER (9.8 vs. 2.1 on COCO and 3.6 vs. 1.9 on LVIS) in terms of the mask AP. This works show stronger performance than the previous SOTA CutLER and SAM on some benchmarks, however, the main results are AR (averaged recall). Recall is important, however, for many downstream tasks, the AP is still the most valuable evaluation metric.

[Unfair comparison] The main baseline CutLER used Cascade Mask RCNN as the segmentation model, while this work used Mask2Former, a stronger segmentation model, as the base model. This makes the performance comparison unfair.

[1] Wang, Xudong, Rohit Girdhar, Stella X. Yu, and Ishan Misra. "Cut and learn for unsupervised object detection and instance segmentation." In Proceedings of the IEEE/CVF Conference on Computer Vision and Pattern Recognition, pp. 3124-3134. 2023.

**Questions:**

My main questions are listed in the weakness section.

---

> ### Author Response · Authors · 2023-11-19
> **Response to Reviewer PuvF**
>
> We appreciate the detailed feedback you provided for our submission. We are encouraged by your acknowledgement of our “unsupervised segmentation” task, “approach to generate unsupervised hierarchical masks,” and paper writing. We provide the following clarifications in response to your concerns:
>
> 1. Technical contributions
>
> - We agree with the reviewer’s perspective, but we would like to point out that such a multi-phase learning paradigm (e.g., discovering pseudo-labels and then learning a detection/segmentation model) is a general pipeline widely-adopted in the existing self-supervised object localization/discovery literature. Indeed, not only does CutLER [R5] (as suggested by the reviewer), but all recent methods also follow this pipeline including LOST [R1], TokenCut [R2], FreeSOLO [R3] (as mentioned by Reviewer vWUT), and FOUND [R4]. That means, the contributions of these different methods do not lie in the introduction of a new pipeline, but instead in proposing unique components to improve this established pipeline for self-supervised object localization/discovery.
>
> - Our SOHES similarly adopts this general multi-phase pipeline, but makes significant contributions through **novel designs for each phase** involved in the learning process, including a global-to-local clustering algorithm for pseudo-label generation, a hierarchical relation learning module, and a teacher-student self-correction phase. In particular, none of the previously mentioned methods [R1-5] consider hierarchical structures among objects/entities, while learning such structures is involved throughout the learning process of SOHES. Given these technical novelties and advancements and our substantial performance improvements over CutLER, we respectfully disagree with the reviewer that the distinctions between our SOHES and CutLER are merely minor implementation differences.
>
> - We appreciate the reviewer’s feedback and will revise the Related Work section and the beginning of the Approach section, to better reflect this general pipeline and our novel components proposed in this work with a hierarchical perspective.
>
> 2. Model performance
>
> We would like to first clarify that, **regarding the open-world segmentation task on MS-COCO/LVIS**, the computed AP number is misleading and cannot accurately reflect the model’s true segmentation performance. **The observed lower AP in our results compared with CutLER’s on MS-COCO/LVIS is a consequence of limitations in the annotations within these datasets**. In contrast, on the SA-1B dataset which mitigates these annotation limitations, our AP (12.9) significantly surpasses CutLER’AP (7.8). Below we explain in detail.
>
> - We chose average recall (AR) over average precision (AP) for the following reason: When evaluating open-world segmentation models (which try to segment “everything”) on datasets with closed-world annotations (such as MS-COCO/LVIS which only include a pre-defined, limited set of entity classes), AP would penalize model predictions that are actually valid entities, but just not annotated by the dataset. Therefore, when the dataset annotations cannot cover all entities, AP becomes misleading for judging the performance of open-world models. Note that this choice of AR over AP is also commonly adopted in the open-world literature. Examples include (but are not limited to) detection [R6], segmentation [R7], and tracking [R8].
>
> - As an example (shown in Table 5), SAM’s AP is lower than CutLER on MS-COCO (6.1 vs. 9.8) and PartImageNet (3.4 vs. 5.3), which definitely cannot imply SAM is a model inferior to CutLER. The lower AP of SAM is due to insufficient annotations in these two datasets, not the capability of SAM. Meanwhile, as the annotated entities increase, AP becomes more and more trustworthy. For instance, AP on SA-1B (the most densely annotated dataset evaluated in our work) matches qualitative comparison and follows the same trend as AR (FreeSOLO < CutLER < SOHES < SAM).
>
> 3. Unfair comparison
>
> - We chose Mask2Former as our segmentation model mainly for making hierarchical predictions. In Cascade Mask R-CNN, each proposal is predicted independently, and therefore, analyzing hierarchical relations between entities would be challenging in Cascade Mask R-CNN. In contrast, the attention modules in Mask2Former allow information exchange among queries, so we can build our ancestor prediction head on Mask2Former.
>
> - For a more comprehensive comparison with CutLER, we train a Cascade Mask R-CNN segmentation model using SOHES without the entity hierarchy. This model still significantly outperforms CutLER with the same segmentation model architecture.
> | Method | Segmentation Model | LVIS | Entity | SA-1B |
> | ------ | ------------------ | ---- | ------ | ----- |
> | CutLER | Cascade Mask R-CNN | 20.2 | 23.1   | 17.0  |
> | SOHES  | Cascade Mask R-CNN | 27.0 | 29.2   | 31.9  |
> | SOHES  | Mask2Former    	| 29.1 | 33.5   | 33.3  |

---

> > ### Author Response · Authors · 2023-11-19
> > **Reference in Response**
> >
> > [R1] Oriane Siméoni, Gilles Puy, Huy V. Vo, Simon Roburin, Spyros Gidaris, Andrei Bursuc, Patrick Pérez, Renaud Marlet, and Jean Ponce. Localizing Objects with Self-Supervised Transformers and no Labels. In BMVC, 2021.
> >
> > [R2] Yangtao Wang, Xi Shen, Shell Xu Hu, Yuan Yuan, James L. Crowley, and Dominique Vaufreydaz. TokenCut: Self-supervised Transformers for Unsupervised Object Discovery using Normalized Cut. In CVPR, 2022.
> >
> > [R3] Xinlong Wang, Zhiding Yu, Shalini De Mello, Jan Kautz, Anima Anandkumar, Chunhua Shen, and Jose M. Alvarez. FreeSOLO: Learning to Segment Objects without Annotations. In CVPR, 2022.
> >
> > [R4] Oriane Siméoni, Chloé Sekkat, Gilles Puy, Antonin Vobecky, Eloi Zablocki and Patrick Pérez. Unsupervised Object Localization: Observing the Background to Discover Objects. In CVPR, 2023.
> >
> > [R5] Xudong Wang, Rohit Girdhar, Stella X. Yu, and Ishan Misra. Cut and Learn for Unsupervised Object Detection and Instance Segmentation. In CVPR, 2023.
> >
> > [R6] Dahun Kim, Tsung-Yi Lin, Anelia Angelova, In So Kweon, and Weicheng Kuo. Learning open-world object proposals without learning to classify. IEEE Robotics and Automation Letters, 7(2): 5453–5460, 2022.
> >
> > [R7] Weiyao Wang, Matt Feiszli, Heng Wang, Jitendra Malik, and Du Tran. Open-world instance segmentation: Exploiting pseudo ground truth from learned pairwise affinity. In CVPR, 2022.
> >
> > [R8] Yang Liu, Idil Esen Zulfikar, Jonathon Luiten, Achal Dave, Deva Ramanan, Bastian Leibe, Aljoša Ošep, and Laura Leal-Taixé. Opening up Open-World Tracking. In CVPR, 2022.

---

> > > ### Author Response · Authors · 2023-11-22
> > > **Any Further Comments?**
> > >
> > > We would like to thank you again for appreciating the task, approach, and results of our work, and for the constructive comments to further improve our paper. Kindly let us know if you have any further comments on our paper, and we would like to do our best to address them in the remaining time.

---

> > > > ### Comment · Reviewer_PuvF · 2023-11-22
> > > >
> > > > Hey authors, thank you for your time in addressing my comments. Most of them have been addressed and I will keep my original rating of borderline accept.

---

> > > > > ### Author Response · Authors · 2023-11-22
> > > > > **Thank you!**
> > > > >
> > > > > Thank you for taking time to reviewing our work and reading our responses. We are glad that our response has addressed your concerns and appreciate your recognition of our paper.

---

### Author Response · Authors · 2023-11-19
**General Response to All**

We appreciate the time and effort invested by all the reviewers in evaluating our manuscript and providing constructive suggestions. In particular, we are grateful for the reviewers’ remarks regarding the strengths of our work:
- (PuvF, vWUT) The target task of our work “holds significant importance,” and “is seldom investigated” by the existing literature.
- (PuvF, vWUT) The approach to generating “hierarchical” entity masks is “pretty interesting” and “new.”
- (PuvF, vWUT, Rono) Experiment results show that our method “significantly outperformed the previous methods”, and even “surpassed SAM in recall on some evaluation benchmarks.”
- (PuvF, Rono) The paper writing is “excellently structured” and “well-articulated, effectively communicating the central ideas.”

We provide detailed clarifications to address the questions raised by the reviewers in the direct responses. We hope our responses can successfully clarify your concerns, and please let us know if further explanation is needed. Once again, we sincerely thank you for the valuable comments and suggestions for improving our work.

---

### Meta-Review · Area_Chair_R3S5 · 2023-12-06

**Metareview:**

All the three reviewers have the borderline acceptance rating after rebuttal (one of them upgraded the rating). The original concerns are: 1) technical contributions of the proposed three-stage method; 2) experimental comparisons for certain settings (e.g., backbone, evaluation metric, sensitivity of hyperparameters); 3) some technical clarity. After the rebuttal, all the reviewers find their concerns mostly resolved. The AC took a close look at the paper and agrees with the reviewers. Hence, the acceptance rating is recommended.

**Justification For Why Not Higher Score:**

It appears that the paper follows the similar steps to existing methods for open-world entity segmentation by revisiting some of the components to improve the performance. Despite that the hierarchical relation learning module is interesting, there are still many feedback needed to incorporate in the final version.

**Justification For Why Not Lower Score:**

N/A

---

### Decision · Program_Chairs · 2024-01-16

Accept (poster)